# CATALYSTBENCH: A COMPREHENSIVE MULTI-TASK BENCHMARK FOR ADVANCING LANGUAGE MODELS IN CATALYSIS SCIENCE

**Xueqing Chen**[1,2,3], **Jian Xu**[2,3,4], **Ludi Wang**[1], **Yang Gao**[5], **Huihan Zhu**[1,2], **Yuanchun Zhou**[1,2,6], **Yi Du**[1,2,6,*], and **Cheng-Lin Liu**[2,3,4,*]

[1]Computer Network Information Center, Chinese Academy of Sciences, Beijing, China
[2]University of Chinese Academy of Sciences, Beijing, China
[3]Zhongguancun Acadamy, Beijing, China
[4]MAIS, Institute of Automation, Chinese Academy of Sciences, Beijing, China
[5]CAS Key Laboratory of Nanosystem and Hierarchical Fabrication, National Center for Nanoscience and Technology (NCNST), Beijing, China
[6]Hangzhou Institute for Advanced Study, University of Chinese Academy of Sciences, Hangzhou, China
xqchen@cnic.cn

## ABSTRACT

The discovery of novel catalytic materials is a cornerstone of chemical engineering and sustainable energy, yet it remains a complex, knowledge-intensive process. While Large Language Models (LLMs) have demonstrated remarkable potential in various scientific domains, their application to catalysis is hindered by the lack of specialized, multi-dimensional benchmarks to guide their development and evaluation. To bridge the critical gap, we introduce CatalystBench, a comprehensive and challenging benchmark meticulously constructed from scientific literature and public datasets, specifically designed to assess the capabilities of LLMs in the nuanced domain of catalyst design. The tasks covered by this benchmark dataset encompass the entire closed-loop process of catalyst development, including reading comprehension, experimental analysis and scheme reasoning. Based on this benchmark, we propose a Multi-head Full-task (MFT) domain-specific fine-tuning method that employs coupling task-specific output heads. We systematically compare with other three distinct fine-tuning strategies: Single-Task (ST), Full-Task (FT) and Multi-head Single-Task (MST). The extensive experiments demonstrate that the MFT strategy consistently achieves the most substantial performance improvements across all tasks, underscoring the effectiveness of explicit multi-task architectures in complex scientific reasoning. The resulting CatalystLLM significantly outperforms a wide array of state-of-the-art open-source and closed-source models on CatalystBench. We will publicly release both the CatalystBench benchmark and the CatalystLLM model, providing the community with a robust evaluation framework and a powerful new tool to accelerate AI-driven research in catalytic materials science.

## 1 INTRODUCTION

The advancement of catalysis is a cornerstone of modern science and industry, pivotal to achieving a sustainable future Swanson et al. (2025); Fu et al. (2025); Zhang et al. (2025b); Song et al. (2025); Ma et al. (2025). In general, the property of catalysts depends on the complex interplay of composition, crystal structure, surface active sites and regulation strategies Zhu et al. (2017); Chen et al. (2020). Designing new catalysts is therefore a formidable challenge: 1) **Vast candidate space**. The combination of multiple chemical elements, possible crystal phases and surface terminations

---

*Yi Du and Cheng-Lin Liu are the corresponding authors.

leads to an astronomical search space, far beyond the reach of exhaustive experimental screening; 2) **Separated knowledge sources**. High-fidelity theoretical datasets derived from density functional theory (DFT) calculations capture key descriptors like adsorption energies and electronic properties Chanussot et al. (2021); Winther et al. (2019), while experimental literature documents synthesis conditions, stability and measured catalytic activities Chen et al. (2024). These two streams are often siloed, with no unified framework linking them for systematic analysis Zhang et al. (2025a). 3) **Lack of realistic evaluation frameworks**. Although AI methods have been applied to materials science, catalysis still lacks a benchmark that reflects the stepwise workflow of catalyst design. The model development remains fragmented and benchmarking across approaches is inconsistent.

Large Language Models (LLMs) have triggered a paradigm shift in "AI for Science" Abramson et al. (2024); Merchant et al. (2023); Szymanski et al. (2023), achieving notable success in domains such as bioengineering Luo et al. (2022); Edwards et al. (2021); Waisberg et al. (2024); Lamb et al. (2024), materials discovery Zhang et al. (2024b); Kristiadi et al. (2024) and industrial process optimization Yang et al. (2023); Saka et al. (2024). However, their deployment in catalytic materials science reveals a critical gap Wang et al. (2025b): existing benchmarks rarely reflect the multi-modal, multi-stage workflows that characterize real catalyst R&D, where precise numerical regression, categorical decision making and open-ended mechanistic reasoning coexist within a single process. We compare the limitations of current benchmarks in the material field across multiple dimensions and conduct a detailed analysis in Appendix A.3. Models trained on homogeneous tasks or unified output formats struggle to preserve accuracy across this spectrum, often suffering loss-landscape interference between qualitatively different objectives.

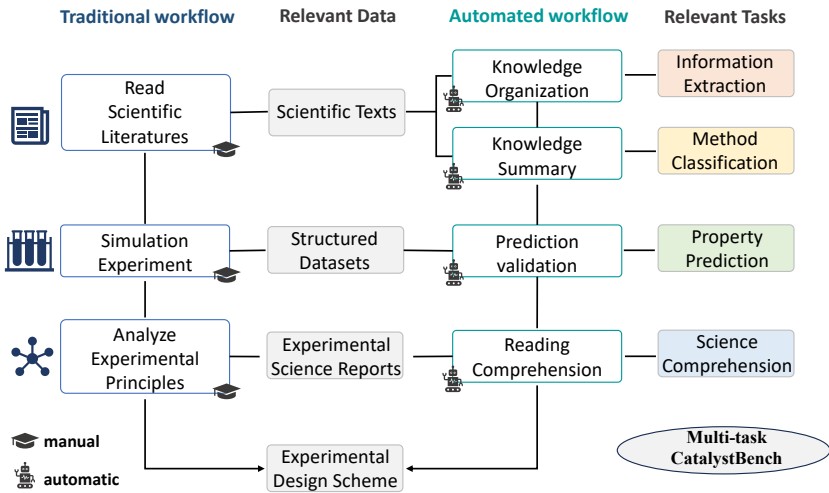

Figure 1: Data-driven model for scientific research processes in the materials field.

To bridge this knowledge gap, we present CatalystBench, the first multi-task benchmark tailored for catalysis science that explicitly unifies high-quality theoretical datasets and curated experimental literature into a structured Q&A format. Beyond aggregating and filtering data, we perform domain-specific curation and transformation so that each evaluation task directly corresponds to a stage in the catalyst design lifecycle, as shown in Fig 1. Given that these tasks span text-based reasoning and numerical property prediction, we propose a Multi-head Full-task Fine-tuning (MFT) strategy. It's an architecture adaptation where classification heads, regression heads and the language modeling head are trained in parallel but decoupled from one another. This architectural decoupling is motivated by the distinctive composition of catalyst design tasks, which uniquely combine high-precision numerical prediction, categorical judgement and open-ended scientific reasoning within a single workflow. While multi-head architectures have been explored in other areas of multi-task learning, this work constitutes the first systematic empirical validation and ablation in the catalysis science domain. We then evaluate our domain-adapted CatalystLLM against strong general and scientific-domain LLM baselines on CatalystBench, finding that it achieves state-of-the-art performance across most task categories. This not only validates the effectiveness of our benchmark and adaptation strategy, but

also yields insights into which aspects of catalyst-related reasoning remain challenging for current LLMs. We highlight the primary contributions of this paper as follows:

- We introduce **CatalystBench**, the first multi-task benchmark for catalysis that integrates theoretical simulation data and experimental literature into a unified and task-oriented format reflecting the actual catalyst design workflow.

- We develop a **domain-specific adaptation strategy**. The Multi-head Full-task Fine-tuning (MFT) approach addresses the heterogeneous nature of catalyst-related tasks by decoupling numerical prediction and language generation within the model architecture. We systematically compare four different fine-tuning strategies and demonstrate the superiority of the MFT strategy.

- We conduct a comprehensive evaluation of CatalystLLM against **multiple strong baseline models**, demonstrating its SOTA performance on our proposed benchmark. Our experiments and detailed analysis not only validate the effectiveness of our tuning strategies but also yield valuable insights into the current strengths and weaknesses of LLMs in catalysis.

## 2 RELATED WORKS

### 2.1 BENCHMARKS IN CHEMISTRY AND MATERIAL SCIENCE

The increasing application of LLMs in scientific discovery has led to the emergence of numerous benchmarks in chemistry and materials science. Initial efforts focused on assessing domain-specific knowledge. For instance, ChemBench evaluate the ability of LLMs on a wide range of chemical knowledge by constructing multiple-choice questions derived from textbooks and expert knowledgeZhang et al. (2024a); Mirza et al. (2024). Similarly, MaScQA Zaki et al. (2024) assesses the understanding of core materials science concepts using graduate-level exam questions, while SCIBENCH Wang et al. (2023b) tests college-level scientific problem-solving abilities.

More recently, benchmarks have evolved to probe more specialized and complex abilities. Chem-CoTBench Li et al. (2025) was specifically designed to evaluate the step-by-step chemical reasoning of LLMs, moving beyond simple factual recall. For predictive tasks, LLM4Mat-Bench Rubungo et al. (2025) provides a comprehensive suite for materials property prediction. Furthermore, Mat-Tools Liu et al. (2025) uniquely evaluates the ability of LLMs to interact with and utilize materials science software libraries. However, existing chemistry-focused LLM benchmarks either emphasize theoretical molecular-level understanding or constrained problem solving and rarely integrate the complementary knowledge from experimental scientific literature Guo et al. (2023); Li et al. (2024); Zhang et al. (2024a); Xie et al. (2024); Chen et al. (2025). This omission is critical for catalysis, where real catalyst surfaces often deviate substantially from the idealized models assumed in theoretical calculations Wang et al. (2023a) and design decisions depend heavily on synthesis conditions, stability data and structure-activity trends reported experimentally. Table 4 compares representative existing chemistry benchmarks with CatalystBench. CatalystBench addresses these gaps by combining high-fidelity theoretical datasets with curated experimental literature in a unified, task-oriented evaluation framework.

### 2.2 DOMAIN-SPECIFIC LLMS FOR CHEMISTRY AND MATERIALS SCIENCE

Beyond evaluating general models, a significant research direction involves creating specialized language models for chemistry and materials science through domain-specific fine-tuning or continued pre-training. Early efforts in this area often involved BERT-style encoder models Chithrananda et al. (2020); Ock et al. (2023); Trewartha et al. (2022); Zhao et al. (2024). More recent studies have shifted toward adaptive adjustments to large-scale generative LLMs. For example, ChemLLM Zhang et al. (2024a) is instruction-tuned on a large set of templated Q&A pairs to handle diverse chemical tasks conversationally and in the materials domain, DARWIN 1.5 Xie et al. (2024) adopts a multi-stage training strategy combining Q&A fine-tuning with multi-task learning to internalize complex materials knowledge. Other generative models include ChemFormer Irwin et al. (2022) for reaction prediction and CrystaLLM Antunes et al. (2024) for generating novel crystal structures.

Recently, several high-impact works further expand the landscape of chemistry LLMs. Llasmol Yu et al. (2024) leverages a high-quality instruction tuning dataset to significantly enhance chemical

reasoning capabilities. Translation between Molecules and Natural Language Edwards et al. (2022) explores bidirectional mapping between molecular structures and textual descriptions, highlighting LLMs' potential for molecular communication. Instructmol Cao et al. (2023a) integrates multimodal data sources to build a versatile, reliable molecular assistant for drug discovery.

However, current domain-specific LLMs are typically optimized for single-task formats or unified output styles, focusing on either symbolic reasoning or purely structural prediction. In contrast, complex scientific scenarios such as catalyst design require the combined ability to comprehend domain literature, reason mechanistically and perform high-precision numerical prediction within a single workflow. The comparison results in Table 5 reveal that most domain-specific large models show limited improvement in numerical prediction tasks, further highlighting the challenges of current multi-task fine-tuning approaches. CatalystLLM tackles this challenge through a multi-head architecture explicitly adapted to the mixed-task nature of catalysis, enabling parallel handling of qualitative and quantitative tasks without mutual interference.

# 3 THE CATALYSTBENCH BENCHMARK

## 3.1 OVERVIEW OF CATALYSTBENCH

In order to explore the abilities of LLMs in the field of materials science, we concentrate on three fundamental capabilities: **Understanding**, **Reasoning** and **Explaining**. Fig 1 illustrates the data-driven paradigm shift in the research process within the field of catalytic materials. In response to this trend, the CatalystBench dataset combines theoretical simulations with scientific experimental data to construct a series of tasks covering the entire process.

## 3.2 CATALYSTBENCH CONSTRUCTION

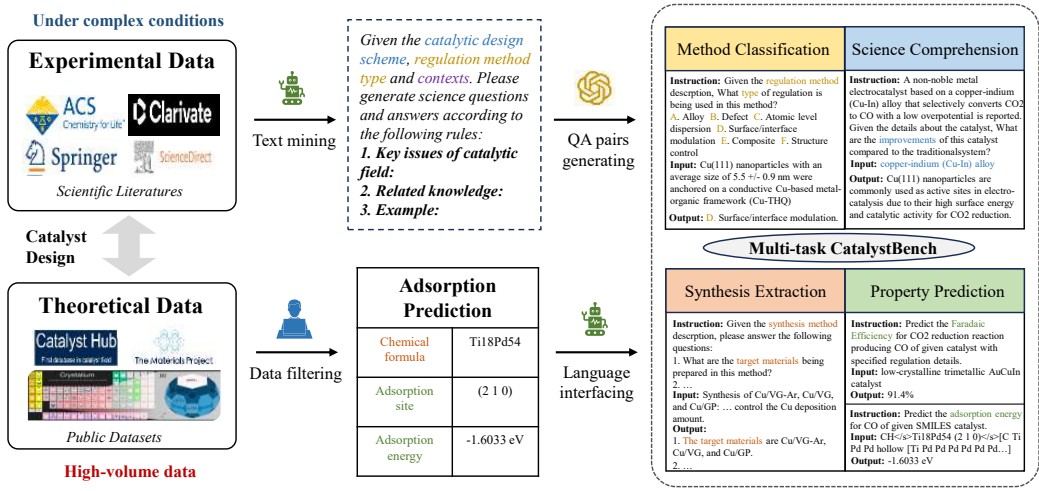

Figure 2: The process for CatalystBench construction and examples demonstration.

### 3.2.1 DATA COLLECTION AND ORGANIZATION

Fig 2 shows the construction process and specific examples of CatalystBench. For theoretical simulation data, we select and integrate 8 publicly available catalytic datasets (Appendix A)to quantitatively evaluate the **Understanding** and **Reasoning** capabilities of LLMs. We filter out key features from descriptors related to catalytic properties in public datasets, such as SMILES strings. We design a set of prompt templates to convert these features and

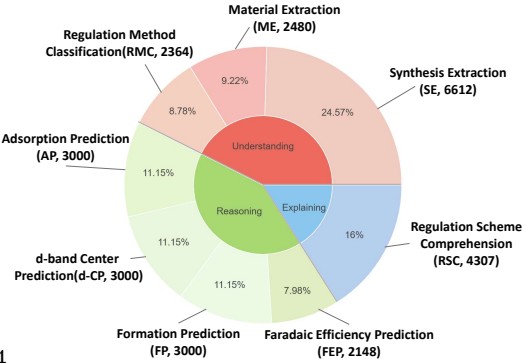

Figure 3: The statistics of CatalystBench Multi-Task Data.

attributes into task-specified natural language
sentences. This leads to instructions that com-
bine inputs and expected outputs. For exam-
ple, the instruction of property prediction like:
'Predict the adsorption energy for CO of given
SMILES catalyst.' with input '$Ti_{18}Pd_{54}$' and
our model should give the output '-1.6033 eV', which can be converted to a numeric value.

Beyond basic understanding and reasoning, we further explore the model's higher-order **Explain-ing** capabilities, including deep understanding of catalytic regulation schemes documented in the literature and accurate grasp of macro-level trends in specific catalytic fields. In this step, we ex-tract catalyst regulation and synthesis schemes from scientific literature based on existing catalytic knowledge frameworks. We use SciQAG Wan et al. (2024) to guide GPT-4o in interpreting a large volume of regulation scheme texts, converting them into high-quality Q&A pairs. Then domain experts perform annotation and filtering. The method details are described in Appendix A.4.

### 3.2.2 CATALYST TASKS

The proportion of these three major categories is illustrated in the Fig 3. We examine these capa-bilities by constructing 8 diverse and broadly acknowledged practical catalytic material tasks. The dataset size for each task in CatalystBench is smaller than that of general benchmarks, reflecting a common constraint in catalytic research: high-fidelity data acquisition is domain-specific and re-quires substantial resources. To avoid data imbalance across tasks, we construct the entire dataset based on the quantity of experimental data, prioritizing task breadth to ensure comprehensive cov-erage of the entire process of the domain of catalyst design.

### 3.3 QUALITY ASSURANCE

To ensure the reliability and reproducibility of CatalystBench, we adopt a multi-tiered quality assur-ance workflow across both theoretically simulated datasets and Q&A pairs generated from experi-mental literature. Automated validation identifies and removes invalid or duplicate entries, followed by expert review of a representative subset. For literature-derived Q&A, a rule-based filtering pro-cess eliminates context questions, while multi-stage generation improves semantic accuracy and diversity. The complete verification procedures and error samples are provided in Appendix A.5.

## 4 METHODOLOGY

### 4.1 BASE LANGUAGE MODEL

We compare the performance of the current open-source models on 3 representative tasks of CatalystBench-full. From Table 5, ChemLLM Zhang et al. (2024a) has bridged this gap through domain-specific instruction fine-tuning on ChemData, infusing the model with a solid chemical foundation. Therefore, we choose ChemLLM-7B as the core base model for multi-task fine-tuning.

### 4.2 FINE-TUNING STRATEGIES

Our goal is to design a fine-tuning architecture that simultaneously harnesses cross-task synergies and avoids performance degradation due to task heterogeneity. Fig 4a illustrates the 4 paradigms investigated, with the Multi-head Full-task(MFT) strategy as our proposed solution.

1) **Single-task:** We fine-tune the LLMs using a training set of each specific task, which allows us to assess the models' ability to adapt to individual tasks and establish a baseline for a fair comparison. We obtain a fine-tuned model for each task in this setup, named 'ChemLLM-ST'.

2) **Multi-head Single-task:** Considering the inherent differences in the attributes of different tasks, we adopt a shared LLM backbone and three task-specific prediction head structures. We obtain a fine-tuned model for each task in this setup, named 'ChemLLM-MST'.

3) **Full-task:** we fine-tune the LLMs using a mixture of all training datasets from 8 tasks, as shown in Fig 4b. The single instruction-following generative head is trained jointly on all 8 CatalystBench

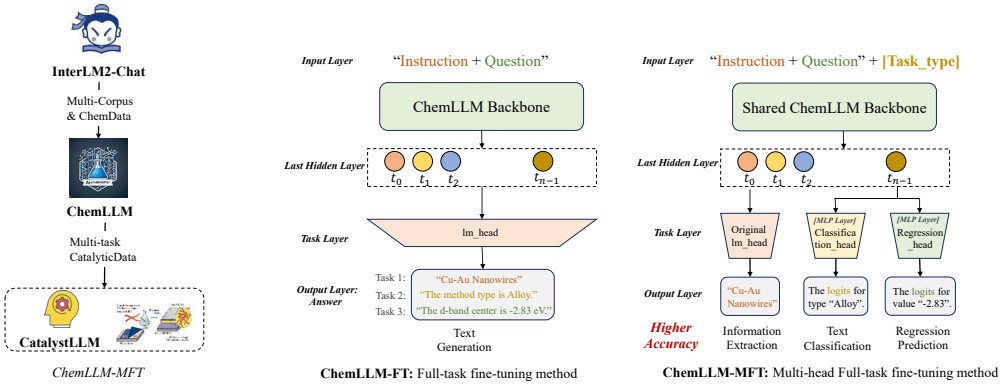

(a) Overview of Cata-lystLLM.

(b) Comparison of model architectures between ChemLLM-FT and ChemLLM-MFT method.

Figure 4: Overview of fine-tuning strategies and final CatalystLLM

tasks. The single instruction follows the generation header (original *lm_head*) during joint training across all 8 CatalystBench tasks, with all task outputs routed to a unified generation decoder. We obtain one fine-tuned model to perform all the tasks in this setup, named 'ChemLLM-FT'.

4) **Multi-head Full-task:** The MFT method extends FT method by decoupling the output space. As shown in Fig 4b, the task type token $\tau$ is appended to the end of the input sequence, where $\tau$ directly controls which dedicated prediction head is activated after the final hidden representation $\mathbf{h}[t_{n-1}]$. Here, $t_{n-1}$ denotes the last non-padding token position in the encoded input, while $\mathbf{h}[t_{n-1}]$ represents the corresponding context embedding generated by the shared ChemLLM backbone Transformer network. For classification tasks, $\mathbf{h}[t_{n-1}]$ is fed into a task-specific MLP classification head, producing a probability distribution over predefined categories. For regression tasks, the same $\mathbf{h}[t_{n-1}]$ input is processed by a regression head optimized for mean squared error, generating a single continuous value. In the original generation task, decoding proceeds directly via the original *lm_head* without architectural modifications, enabling the model to generate domain-specific explanatory text.

By training the shared backbone jointly across all tasks while isolating output modalities, MFT method alleviates the interference between loss landscapes of qualitatively different tasks. In practice, we balance their contributions via a weighted composite loss and compare the impact of loss-function in Appendix B.6. This approach leverages cross-domain feature learning that is critical in catalytic material research, while preserving the precision of task-specific inference.

# 5 EXPERIMENTAL SETUP

## 5.1 BASELINE MODELS

For the sake of fairness, We select 15% of the data for each task in CatalystBench as the test set, according to catalyst material type. Then we compare the performance of different LLMs on CatalystBench-test, including open-source models of comparable scale, such as LLaMA-2 Touvron et al. (2023), Mistral Jiang et al. (2024), ChatGLM GLM et al. (2024) and Qwen3-8B Bai et al. (2023), as well as closed-source models with strong instruction-following capabilities, such as GPT-3.5 Ye et al. (2023), GPT-4 Achiam et al. (2023) and deepseek. Additionally, for regression tasks involving the prediction of material properties, we introduce comparisons with competitive ML algorithms. We select some SOTA baseline models, such as CatBERTa Ock et al. (2023) and GAP-CatBERTa Ock et al. (2024) from OC20 dataset and ML algorithm such as GPTchem Jablonka et al. (2024). For certain tasks, we directly use the ML results from the original dataset papers Gao et al. (2023). We provide a detailed introduction of these baseline models in Appendix B.

## 5.2 FINE-TUNING SETUP

To achieve efficient fine-tuning, we adopt Low-Rank Adaptation (LoRA) to reduce the computational cost with a rank of 8, a scale factor of 16.0 and a dropout rate of 0.1. We fine-tune all linear modules. The training uses the AdamW optimizer with a learning rate of 5e-5, combined with a linear decay scheduler with warm-up. Additionally, we further enhance training speed and model robustness through techniques such as bf16 mixed precision and Flash Attention-2.

## 5.3 EVALUATION PROTOCOL

We employ a suite of tailored evaluation metrics. For the Text Classification and Information Extraction tasks, we report both Accuracy and the balanced F1-score to account for potential class imbalances. For the Regression tasks, we use MAE and $R^2$ score to measure predictive accuracy.

Given the non-factual features of the semantic Q&A task, we develop a multi-dimensional evaluation protocol. This protocol begins by calculating the STS score, which measures the sentence-level semantic similarity between the generated answer and the reference answer. To assess the domain expertise of the answers, we employ a model-based evaluation framework M. Bran et al. (2024) where both gpt-4o and deepseek-r1 are prompted to score the generated answers on a scale from 1-10 across three criteria: reasonableness, accuracy and Usability. Finally, we conduct human evaluation on a random subset of the test dataset, with domain experts providing the final judgment on the model's practical applicability and domain-specific correctness(Details in Appendix C).

## 6 RESULTS AND ANALYSIS

## 6.1 COMPARISON OF FINE-TUNING STRATEGIES

After fine-tuning the ChemLLM-7B model, we evaluate their performance on CatalystBench-test for each task. In Fig 5, we compare task performance to identify how different fine-tuning strategies influence the results. The results show that the MFT strategy achieves the best performance, with the highest average performance improvement rate relative to the ST baseline, reaching 12.44%. This indicates that combining multi-task learning with task-specific output heads is more effective than applying either technique in isolation. Among these, MT fine-tuning resulted in an average improvement of 9.24%, while task-specific output head design achieve an average improvement of only 5.13%. This highlights that coupling architectures help tailor the model's outputs for a specific task format, but the impact is limited without multi-task train-

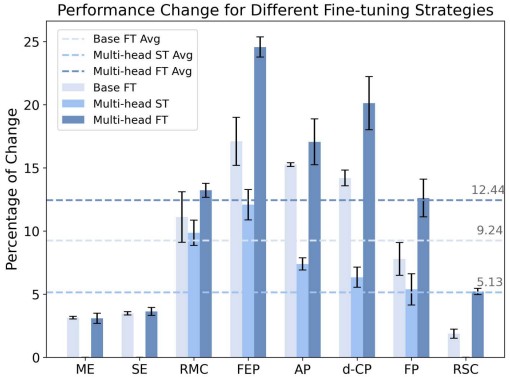

Figure 5: Comparison of the impact of different fine-tuning strategies on task performance. Base-ST serves as the baseline model and the bar chart shows the performance differences between different models on specific tasks.

ing. When all tasks share underlying semantic features, the model can leverage these features to learn shared and transferable representations. We name the final ChemLLM-MFT model as CatalystLLM, an open-source foundational LLM specifically designed for catalytic science. We provide a detailed analysis of the suitability of MFT architecture for catalytic process tasks in Appendix B.1.

## 6.2 MAIN BENCHMARK RESULTS

We evaluate the performance of LLMs on catalytic tasks on CatalystBench-test and report the results in Table 1 and Table 2. Table 1 compares the performance of CatalystLLM with other LLMs on tasks with factual answers. The results show that CatalystLLM achieve the best performance across all 7 tasks and 12 metrics, significantly outperforming general LLMs of similar scale. Compared to ChemLLM, CatalystLLM demonstrates a significant improvement in its capabilities to understand and apply catalytic knowledge, highlighting the effectiveness of multi-task catalytic data fine-tuning.

| Model | ME | | SE | | RMC | | FEP | | AP | | d-CP | | FP | |
|---|---|---|---|---|---|---|---|---|---|---|---|---|---|---|
| | ACC | F1 | ACC | F1 | ACC | F1 | R2 | MAE | R2 | MAE | R2 | MAE | R2 | MAE |
| **Closed-source LLMs** | | | | | | | | | | | | | | |
| claude-3 | 0.93 | 0.95 | 0.84 | 0.90 | 0.58 | 0.72 | 0.30 | 6.73 | 0.37 | 3.58 | 0.33 | 4.21 | 0.39 | 5.24 |
| Gemini-2.5 | 0.97 | 0.98 | 0.87 | 0.91 | 0.66 | 0.80 | 0.29 | 3.19 | 0.34 | 4.15 | 0.36 | 5.78 | 0.40 | 4.98 |
| gpt-3.5 | 0.89 | 0.93 | 0.78 | 0.86 | 0.49 | 0.62 | 0.28 | 5.25 | 0.35 | 4.20 | 0.34 | 4.50 | 0.38 | 5.41 |
| gpt-4o | 0.96 | 0.98 | 0.88 | 0.92 | 0.69 | 0.81 | 0.36 | 3.10 | 0.43 | 3.05 | 0.41 | 3.35 | 0.46 | 4.02 |
| gpt-4.1 | **0.98** | **0.99** | **0.91** | **0.95** | 0.75 | 0.85 | 0.56 | 2.51 | 0.61 | 2.40 | 0.59 | 2.15 | 0.65 | 2.68 |
| **Open-source LLMs** | | | | | | | | | | | | | | |
| deepseek-v3 | 0.92 | 0.95 | 0.83 | 0.89 | 0.61 | 0.74 | 0.33 | 4.20 | 0.40 | 3.55 | 0.38 | 3.85 | 0.43 | 4.45 |
| LLaMA2-7B | 0.83 | 0.86 | 0.73 | 0.79 | 0.39 | 0.50 | 0.25 | 4.47 | 0.24 | 4.63 | 0.34 | 3.79 | 0.35 | 4.12 |
| Qwen3-8B | 0.86 | 0.91 | 0.74 | 0.83 | 0.42 | 0.54 | 0.26 | 4.59 | 0.31 | 3.85 | 0.32 | 3.49 | 0.36 | 4.26 |
| Mistral-7B | 0.88 | 0.92 | 0.76 | 0.84 | 0.44 | 0.57 | 0.29 | 3.09 | 0.37 | 3.49 | 0.36 | 4.14 | 0.39 | 5.15 |
| ChatGLM3-6B | 0.84 | 0.90 | 0.74 | 0.83 | 0.45 | 0.57 | 0.33 | 4.87 | 0.39 | 3.07 | 0.37 | 3.48 | 0.41 | 4.38 |
| ChemLLM | 0.93 | 0.95 | 0.79 | 0.88 | 0.52 | 0.66 | 0.45 | 2.80 | 0.63 | 2.05 | 0.54 | 2.36 | 0.64 | 2.75 |
| Darwin1.5 | 0.91 | 0.94 | 0.79 | 0.89 | 0.50 | 0.64 | 0.44 | 2.81 | 0.59 | 3.13 | 0.54 | 2.42 | 0.68 | 2.01 |
| **CatalystLLM** | **0.98** | **0.99** | 0.89 | 0.94 | **0.81** | **0.89** | **0.73** | **1.72** | **0.81** | **1.24** | **0.73** | **1.49** | **0.80** | **1.34** |

Table 1: The performance of CatalystLLM with other LLMs on tasks with factual answers. The best model is in bold font and the second-best is underlined.

ChemLLM and Darwin1.5 outperform top general models on most metrics. This indicates that specialized models can learn more precise chemical named entities, reaction conditions and critical features in complex structure-activity relationships through fine-tuning in specific domains.

Notably, general LLMs perform exceptionally well in tasks such as ME, SE and RMC tasks. For instance, gpt-4.1 achieves higher accuracy and F1-scores than CatalystLLM in the SE task. However, in numerical regression prediction tasks, there is a significant gap in capability between general models and specialized models, indicating that general LLMs currently can not accurately perform prediction tasks which require deep scientific reasoning and quantitative calculations. For numerical regression tasks, we also compare the results of CatalystLLM and traditional single-task competitive ML algorithms in Appendix B.5.

Table 2 compares the performance of CatalystLLM with other LLMs in semantic understanding tasks, including multi-dimensional evaluation metrics. In all three evaluation dimensions, CatalystLLM achieves the optimal or suboptimal results, demonstrating the potential of domain-specific fine-tuning to enhance answer quality. Additionally, the STS scores of general LLMs are typically higher than those of open-source models, reflecting the superior answer generation capabilities. General LLMs typically have higher LLM Scores, while specialized models including CatalystLLM have relatively higher expert scores. This reflects two phenomenons: 1) Top general models generate answers that are fluent, logically clear and comprehensive, which aligns with the preferences of gpt-4o and deepseek-r1; 2) General models are prone to scientific hallucinations in specialized fields like catalytic science Xu et al. (2024). These answers may contain incorrect catalyst performance analyses or explanations that conflict with physical chemistry principles and such errors can only be identified by domain experts. We provide an error example of LLM-score and expert scores in Appendix C.2. Additionally, we provide case studies to highlight the impact of CatalystLLM's domain expertise on catalytic materials scientists in Appendix C.3.

| Model | STS | LLM Scores | | Experts @100 |
|---|---|---|---|---|
| | Scores | gpt-4o | deepseek-r1 | Score |
| **Closed-source LLMs** | | | | |
| claude-3 | 0.73 | 0.74 | 0.73 | 0.46 |
| Gemini-2.5 | 0.74 | 0.75 | 0.76 | 0.47 |
| gpt-3.5 | 0.71 | 0.69 | 0.71 | 0.41 |
| gpt-4o | 0.73 | 0.71 | 0.72 | 0.46 |
| gpt-4.1 | 0.72 | **0.85** | **0.88** | 0.56 |
| **Open-source LLMs** | | | | |
| deepseek-v3 | 0.73 | 0.84 | 0.86 | 0.57 |
| LLaMA2-7B | 0.63 | 0.57 | 0.52 | 0.39 |
| Qwen3-8B | 0.62 | 0.70 | 0.72 | 0.47 |
| Mistral-7B | 0.66 | 0.68 | 0.42 | 0.42 |
| ChatGLM3-6B | 0.61 | 0.58 | 0.60 | 0.38 |
| ChemLLM | 0.68 | 0.64 | 0.68 | 0.54 |
| Darwin1.5 | 0.67 | 0.59 | 0.61 | 0.49 |
| **CatalystLLM** | **0.79** | 0.82 | 0.86 | **0.75** |

Table 2: The performance of CatalystLLM with other LLMs in semantic understanding tasks. The best model is in bold font and the second-best is underlined.

## 6.3 ABLATION STUDY

We compare the impact of different dataset settings and experimental settings on model capabilities to explore the key factors that determine LLM domain capabilities. In the experimental setup, all models are based on the optimal MFT architecture.

### 6.3.1 COLLABORATIVE EFFECTS OF TASK COMBINATIONS

Figure 6: The experiments on collaborative effects of task combinations.

We investigate the potential collaborative relationship among different types of tasks under multi-task learning framework. We remove regression tasks, classification tasks and information extraction tasks from the training data separately and then evaluate the corresponding performance. As shown in Fig 6, compared to the Full-MFT baseline, the performance of all ablation models decrease on the corresponding tasks. Additionally, we divide these tasks into two groups based on the type of input data: Text data and Chemical data and compare the collaborative effects of tasks within and between groups.

**Intra-group tasks exhibit strong synergistic effects.** For example, when RMC data is removed, the performance of other tasks within the same group decreases significantly. This indicates that learning "how to regulate" (RMC) provides critical contextual knowledge for LLM to understand "how effective the regulation is" (FEP) and "the significance of the regulation scheme" (RSC).

**Inter-group tasks exist certain degree of synergistic effects.** For example, in the "MFT w/o Regression" setting, the performance of the RMC and RSC tasks is also affected to some extent. This reveals a deeper phenomenon of domain-specific fine-tuning model: it does not merely memorize input-output patterns from training data but construct an overall knowledge model of the catalytic domain. CatalystLLM can span different data modalities, such as chemical formulas and natural language, thereby aiding decision-making in other tasks.

### 6.3.2 THE IMPACT OF INPUT FORMAT ON MODEL PERFORMANCE

To investigate the advantages of Catalyst-Bench, we investigate the impact of input data format on the performance of LLMs. The experiments aim to compare two input strategies: one provided only unstructured core information, while the other provided structured and complete input. The two input strategies and related generated prompts are shown in the Appendix E.2. As shown in Fig 7, given the same amount of information, the quality and format of input data are key factors determining the upper limit of LLMs' performance in professional tasks, especially in the chemical representation of catalysts. The single SMILES string sequence can only provide a macro-level representation, while structured knowledge allows the model to learn the quantity and connection methods of internal atoms, thereby learning potential structure-activity relationships.

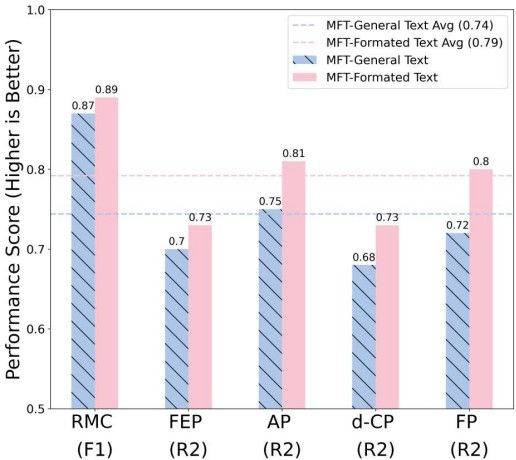

Figure 7: The impact of input format on model performance.

## 7 CONCLUSION AND DISCUSSION

This work presents CatalystBench, a comprehensive benchmark dataset covering crucial processes in catalytic science and systematically demonstrates the effectiveness of multi-head full-task fine-tuning in adapting to complex scientific tasks. Building upon this foundation, we develop CatalystLLM, a high-performance specialized language model tailored for the field of catalysis. Our experiments demonstrate the superiority of CatalystLLM over existing LLM models and highlight the pivotal role of CatalystBench in enhancing performance. Further analytical experiments also provide crucial insights for developing LLM models for catalytic materials.

CatalystLLM attains SOTA performance, yet certain limitations remain. Its knowledge scope is constrained by the coverage of the CatalystBench dataset, potentially reducing predictive accuracy for catalytic materials outside this range. While the 7B-parameter scale balances efficiency and capability, larger models may exhibit superior reasoning and learning capacity. Furthermore, the inclusion of data from existing literature introduces potential bias related to source selection. Future work will aim to broaden CatalystBench to encompass a wider variety of catalytic systems and tasks, enhance CatalystLLM's downstream inference and embed the model within a closed-loop catalyst discovery framework, enabling AI predictions to directly inform synthesis and characterization, thereby accelerating the development of novel materials.

## ETHICS COMPLIANCE

This paper adheres to the ICLR Code of Ethics. We acknowledge that our research follows ethical guidelines, ensuring compliance with all relevant regulations. No human subjects were involved in this research. All data used in the study are publicly available datasets or generated synthetically. We have taken care to ensure that all data used complies with relevant data protection laws. Datasets used in this research do not contain any personal, confidential, or sensitive information. We confirm that the research presented is original and has not been plagiarized. All sources and data are properly cited and documented. We confirm that the research complies with all applicable laws and ethical standards, including data usage and research practices. This statement is meant to address potential concerns in accordance with the ICLR Code of Ethics.

## REPRODUCIBILITY STATEMENT

We ensure reproducibility by providing all necessary resources for others to replicate our experiments. Specifically:

1) Benchmark Data and Experiment Setup: All datasets used in experiments and detailed data processing steps are publicly accessible. The experimental setup is also provided, including hyperparameters, evaluation metrics and the benchmarking environment.

2) Source code: The code used for data processing and benchmarking experiments is provided in the supplementary materials, containing all scripts required to reproduce the experiments described in the paper.

By providing these resources, we aim to achieve full reproducibility of our research findings and facilitate subsequent studies based on this work. Detailed information required for reproduction can be found in the supplementary materials.

## ACKNOWLEDGEMENTS

This work are supported by Zhongguancun Academy Project No.02012501, in part by the Natural Science Foundation of China under Grant No.T2322027 and 62442204, in part by the Key Research Program of the Chinese Academy of Sciences under Grant No.RCJJ-145-24-20, partially by the Natural Science Foundation of China(No.92470204) and partially by Young Scientists Fund Category C (No.62506357).

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

TABLE OF CONTENTS IN APPENDIX

## A DATA CONSTRUCTION

We introduce the various sources of CatalystBench data, describe the specific tasks within each category and clarify the complete process used to compile this dataset.

| Ability | Task | Task Type | Datasets | Dataset Type | #count | Evaluation Metrics |
|---|---|---|---|---|---|---|
| Understanding | Material Extraction (ME) | Generation | ElectroCatalytic Reduction(1 & 2) | Experimental Data | 2,480 | Accuracy, F1 score |
| | Synthesis Extraction (SE) | Generation | ElectroCatalytic Reduction(1 & 2), Solution-based synthesis dataset | Experimental Data | 6,612 | |
| | Regulation Method Classification(RMC) | Classification | ElectroCatalytic Reduction(1 & 2) | Experimental Data | 2,364 | |
| Reasoning | Faradaic Efficiency Prediction(FEP) | Prediction | ElectroCatalytic Reduction(1 & 2) | Experimental Data | 2,148 | $R^2$, MAE |
| | Adsorption Prediction (AP) | Prediction | OC20-Dense, SACs Dataset | Theoretical Simulation | 3,000 | |
| | d-band Center Prediction(d-CP) | Prediction | SACs Dataset, Catalysis-Hub, Catalytic Material Database(CMD) | Theoretical Simulation | 3,000 | |
| | Formation Prediction (FP) | Prediction | SACs Dataset, Material Project, Catalytic Material Database(CMD) | Theoretical Simulation | 3,000 | |
| Explaining | Regulation Scheme Comprehension(RSC) | Generation | ElectroCatalytic Reduction(1 & 2) | Experimental Data | 4,307 | STScore, LLM Score |

Table 3: The statistics of all tasks, datasets, dataset type and evaluation metrics.

### A.1 DATA SOURCE

We extract data from 8 open-source databases. The name and description of each dataset are listed below.

- **ElectroCatalytic Reduction 1 Wang et al. (2023a)**[*]: An open-source corpus of electro-catalytic CO2 reduction extracted from science literatures Du et al. (2023).

- **ElectroCatalytic Reduction 2 Chen et al. (2024)**[†]: A text-mining dataset describing the CO2 reduction process catalyzed by copper-based electrocatalysts Wang et al. (2025a), which specifically includes material, regulation method, product, Faradaic efficiency and relevant synthesis conditions.

- **Solution-based synthesis dataset Kononova et al. (2019)**[‡]: A dataset of 35,675 solution-based synthesis procedures extracted from the scientific literature. Each procedure contains essential synthesis information including the precursors and target materials, their quantities and the synthesis actions and corresponding attributes.

- **OC20-Dense Chanussot et al. (2021)**[§]: The largest dataset of catalyst-adsorbate interactions to date, designed to accelerate catalyst discovery through machine learning. The dataset contains over a million DFT calculations covering an extremely wide range of catalyst materials (including bulk, surfaces and nanoparticles) and adsorbates.

- **Catalytic Material Database(CMD)**[¶]: CMD contains material composition, properties, reactions, products and other information.

- **SACs Hiragond et al. (2022)**[‖]: A dataset of catalyst samples constructed for electrocatalytic reactions such as OER or ORR. SACs contains dozens of transition metals, combined

with several different coordination environments and substrates, resulting in hundreds of catalyst samples.

- **Catalyst Hub Winther et al. (2019)**[**]: A featured database for surface reactions contains more than 100,000 chemisorption and reaction energies obtained from electronic structure calculations and is continuously being updated with new datasets.

- **Materials Project Jain et al. (2013)**[††]: The Materials Project provides computed information on known and predicted materials as well as powerful analysis tools to inspire and design novel materials.

## A.2 TASK FORMULATION

We evaluate the understanding, reasoning and explanation capabilities of LLMs in catalyst design by constructing 8 widely recognized practical catalytic material tasks. Table 3 summarizes these tasks, including their task types from a machine learning perspective, the datasets used for evaluation and the evaluation metrics.

**Simple Instruction Data Synthesis Method:** Converting structured chemical data into reasoning-adjusted data suitable for training LLMs presents two main challenges: 1) the creation of diverse templates and 2) the integration of chemical logic and reasoning into QA pairs. The information extraction tasks focus on identifying composition and operational parameters related to material design from scientific literatures. The classification tasks predict discrete categories or labels, while the regression tasks focus on continuous numerical property values. For template diversity, we initially developed a foundational seed template to meet the requirements of specific tasks. Using gpt-4o, We generate a series of prompt templates that express different meanings while maintaining semantic consistency. These diverse templates enhance the model's ability to interpret and respond to different instruction formats. For each structured data entry, we randomly select one of these templates to create a single-round dialogue sample.

The source datasets typically use various representations of catalytic materials to characterize the corresponding catalytic properties Cao et al. (2023b); Balaji et al. (2023); Xu et al. (2023); Ock et al. (2023). We propose a key hypothesis: When the model is only provided with the material common name and its SMILES representation, it is difficult for LLM to learn the complex structure-property relationships implicit in the data. To validate this hypothesis and explore the impact of input information richness on model performance, we create prompts based on the source dataset to generate different types of catalytic characterization input data. The prompts are included in Appendix E.2.

**Complex Scenario Data Synthesis Method:** For information-rich complex texts such as synthesis schemes, we enriched the data by constructing multi-round dialogues to provide contextual depth and logical consistency. By simulating the deep thinking and step-by-step reasoning involved in catalytic material design processes, we created a highly specialized multi-round dialogue dataset, reducing the need for dialogue rounds and human intervention. The above template examples are included in Appendix D.

## A.3 THE COMPARISON OF CHEMICAL BENCHMARK

Catalysis research presents unique challenges that distinguish it from broader chemistry or materials science applications, particularly in the combination of heterogeneous knowledge modalities and the multi-stage, interdependent workflow of real catalyst design. While prior LLM benchmarks in chemistry and materials science have explored domain-specific reasoning or property prediction,

---

[*]https://doi.org/10.57760/sciencedb.07106
[†]https://doi.org/10.57760/sciencedb.13290
[‡]https://doi.org/10.6084/m9.figshare.16583387.v4
[§]https://opencatalystproject.org/
[¶]http://cmd.us.edu.pl/catalog/
[‖]https://catalysis-ncepu-hvkydg736ykqeq26d5gxrn.streamlit.app/
[**]http://www.catalysthub.net/
[††]https://next-gen.materialsproject.org/

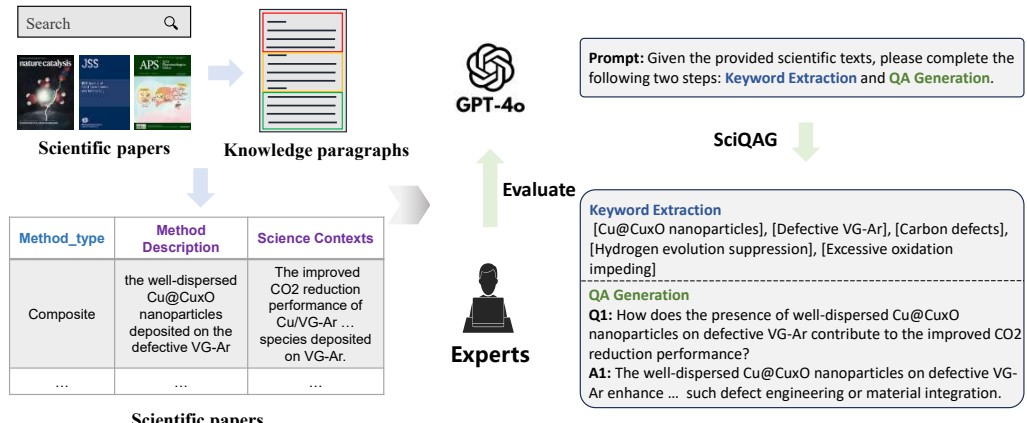

Figure 8: The process of converting regulation method text into high-quality Q&A pairs using SciQAG.

none have simultaneously aligned tasks to the full, practical catalyst R&D pipeline — from extraction of synthesis parameters in literature, through structure–property prediction, to the mechanistic interpretation of regulation strategies.

| Benchmark | Domain Focus | Task Type | Data Modalities | Workflow Alignment to Real Catalyst Design |
|---|---|---|---|---|
| ChemBench | General chemistry knowledge | Multiple-choice factual recall | Text only | × Fragmented |
| MaScQA | Materials science concepts | Q&A from graduate-level examinations | Text only | × Fragmented |
| SCIBENCH | College-level scientific problem solving | Multi-domain Q&A | Text only | × Fragmented |
| ChemCoTBench | Chemical reasoning | Step-wise chain-of-thought | Text symbolic expressions | × Fragmented |
| LLM4Mat-Bench | Material property prediction | Regression/classification on numerical properties | Structured material descriptors | × Partial |
| CatalystBench (ours) | Catalysis-specific, combining theory experiment | 8 tasks: information extraction, method classification, numerical property prediction, regulation scheme comprehension | Text structured chemical descriptors numerical values | ✓ Full workflow coverage |

Table 4: The comparison of CatalystBench and the current Benchmark across dimensions such as domain categories, data modalities and task types.

Table 4 compares representative existing chemical benchmarks with CatalystBench in terms of task categories, data modalities and alignment to real-world catalyst design workflows. The comparison illustrates that CatalystBench uniquely integrates both high-fidelity theoretical simulations and curated experimental literature into a task suite directly reflecting the sequential steps of modern catalytic materials discovery.

## A.4 Q&A GENERATION

We use SciQAG Wan et al. (2024) to guide GPT-4o in interpreting a large volume of regulation scheme texts, converting them into high-quality comprehension Q&A pairs. The pre-defined task is defined as follows: Given a seed input catalytic design scheme text $T$, for each input text $t$, the generator should first generate 5 keywords $k$ that capture the most important terms and concepts in the text and then generate a set $S = \{(q_i, a_i)\}_{i=1}^{n}$ focusing on the generated keywords $k$, where $\forall i \in \{1, 2, \ldots, 5\}$, $q_i$ is the question and $a_i$ is the answer to $q_i$. We use the 1,198 catalytic design scheme texts from the ElectroCatalytic Reduction dataset as input to generate 5,990 seed Q&A pairs by prompting GPT-4o. The generating prompts are included in Appendix E.1. To reduce the occurrence of non-knowledge-based questions that can only be answered using information from the given article, we develop a rule-based method to remove all pairs containing "this paper" or "this study." As shown in Fig 8, we ultimately generate 4,307 semantic understanding text Q&A pairs in this step.

## A.5 QUALITY ASSURANCE DETAILS

In order to guarantee the scientific credibility of CatalystBench and to ensure reproducibility of results, a multi-layered quality assurance framework was implemented during dataset development. This framework covers both theoretically simulated catalytic property data and open-ended Q&A content extracted and generated from experimental literature.

### A.5.1 THEORETICAL Q&A DATA CALIBRATION

For task data derived from theoretical simulations, we first perform foundational consistency checks before feeding it into the conversion interface and generating corresponding QA-formatted content. This process removes samples with obvious errors, such as invalid SMILES representations, missing chemical formulas, or attribute values falling outside the original data's statistical range. Simultaneously, samples with duplicate structures or identical text are cleaned using hash fingerprint comparison and field uniqueness checks. To maintain scientific rigor while preserving large-scale data volume, we randomly selected 20% of samples from the cleaned theoretical dataset and invited two independent experts in catalytic materials for manual verification. The review process employs a double-blind cross-evaluation approach to ensure conclusions are not influenced by individual reviewer preferences and to achieve explicit consensus on answer consistency assessments.

### A.5.2 OPEN-ENDED Q&A DATA CALIBRATION

For open-ended comprehension Q&A data sourced from experimental literature and automatically generated by SciQAG-guided GPT-4o, we implement an additional rule-based filtering system to remove potentially noisy article-internal reference questions. This includes questions relying solely on non-generic knowledge cues like "this paper" or "this study." This filtering effectively mitigates the risk of model over-reliance on specific textual contexts during training. Furthermore, during the generation phase, we avoid a single-round direct instruction-to-output approach. Instead, we design a multi-round generation process: first extracting keywords and key mechanisms, then guiding the model to generate Q&A focused on core scientific concepts, followed by structured rewriting and semantic consistency review. This multi-round strategy significantly mitigates GPT-4o's tendency toward fixed stylistic expressions and non-scientific rhetoric in lengthy responses. Consequently, the retained data exhibits greater expressive diversity, enhanced scientific rigor and high consistency with actual catalytic mechanisms.

We further introduce a verifiable knowledge attribution mechanism to reduce and identify potential hallucination content. During the extraction of contextual keywords, we concurrently build an internal retrieval database comprising three elements: 1) We retain the unique DOI identifiers of corresponding literature to ensure each Q&A pair can be traced back to its original publication; 2) We preserve the complete contextual paragraphs underlying generated questions for offline review and revalidation; 3) We associate extracted core scientific keywords with their corresponding text fragments, establishing a keyword-context-DOI mapping. This structured database not only provides an additional quality control measure beyond expert review for open-ended Q&A data but also enables rapid programmatic verification of answer consistency with original sources. This maintains the high standards for scientific verifiability and reproducibility of our datasets.

We also randomly sample 50% of the dataset and conduct manual verification by experts in the field of catalysis. We invite several experts from the National Nanotechnology Center's catalysis field to participate, including 2 associate researcher and 6 doctoral student. The review process covers three central aspects: **1) The reasonableness and scientific value of the open-ended Q&A task questions**, whether they reflect the development trends and focal points of the catalysis field; **2) The accuracy and completeness of the answers**, ensuring they include all key conditions; and **3) The relevance and contextual fidelity of the Q&A tasks**, ensuring there are no hallucinations generated by the model itself. After an initial filtering, approximately 90.8% of the Q&A tasks are identified accurate and scientifically meaningful. To further validate the rigor of our filtering process, we conduct IAA analysis on 500 randomly selected samples. Two domain experts independently annotate each sample as accept or reject achieving a Cohen's Kappa coefficient of 0.75. This further confirms that the dataset reflects domain knowledge rather than model bias.

A.6   ERROR SAMPLES

The errors in automatically generated open-ended Q&A responses primarily stem from models introducing inferences inconsistent with mainstream consensus in the catalysis field, mismatched experimental conditions, or unfounded explanations of reaction mechanisms during answer generation. To clarify the feature of this issue, we present several representative error samples, all annotated and verified by two independent domain experts.

---

**Error Sample 1**

**Question:**
What is the specific role of Zn sites in the CO electro-reduction process within this Cu-Zn bimetallic catalytic system?

**Reference Answer:**
The Zn site serves as an efficient active center for CO generation, selectively reducing CO to the *CO intermediate and synergistically promoting subsequent C-C coupling with the Cu site to form ethylene.

**The answer generated by LLM:**
Zn sites directly participate in C-O bond cleavage, promoting the reduction of CO to CO while enhancing the selectivity of Cu sites toward ethylene.

**The error causes:**
The model erroneously interprets the Zn site's action as directly participating in bond cleavage, which contradicts the explicitly documented mechanism in the literature of "indirect action via intermediate generation." This constitutes a mechanistic error.

---

---

**Error Sample 2**

**Question:**
What is the primary contribution of nitrogen doping when employing an electrocatalytic system supported by nitrogen-doped graphene for Fe single-atom active sites?

**Reference Answer:**
Nitrogen doping alters the local electronic structure, increases the electron density at active sites and improves reaction kinetics by enhancing adsorption capacity.

**The answer generated by LLM:**
Nitrogen doping enhances the carrier conductivity, enabling it to maintain structural stability under high current densities.

**The error causes:**
The generated answer unduly emphasizes enhanced conductivity while overlooking the dominant effects verified in the literature: regulation of active site electronic structures and improvement in adsorption capacity. This error constitutes a critical omission of key factors.

---

**Error Sample 3**

**Question:**
What is the primary purpose of introducing nanoporous structures in Au-Cu alloys for CO electrolytic reduction?

**Reference Answer:**
The porous structure increases the specific surface area and the number of exposed active sites while improving the mass transfer efficiency of reactants.

**The answer generated by LLM:**
Naphthalene-based structures are primarily employed to suppress the formation of methane byproducts, enhancing selectivity through altered electron cloud distribution.

**The error causes:**
The response focuses solely on selective enhancement and methane suppression as its sole objectives, lacking descriptions of surface structure and mass transfer effects. This results in insufficient scientific rationale and inconsistency with the original context keywords.

---

## A.7 DATA DISTRIBUTION AND DIVERSITY ANALYSIS

To comprehensively demonstrate the composition of CatalystBench, this section conducts detailed statistical and diversity analysis of the dataset. We aim to reveal its coverage breadth and data distribution characteristics across dimensions such as catalyst types, regulation method types and reaction types through quantitative data, thereby validating its rationale and challenge as a comprehensive benchmark. Fig 9 illustrates the distribution characteristics of the catalysts and reaction types involved in CatalystBench.

**Distribution of Key Metal Elements:** We compile the key metal elements in catalysts across the dataset, particularly in tasks derived from experimental literature such as ME, SE and RMC tasks. As shown in Fig 9a, $Cu$ is the most frequently occurring element, accounting for 43.7%. This aligns closely with our data sources' focus on the current state of research in $CO_2$ electrocatalytic reduction($CO_2RR$), where copper-based catalysts represent a major research focus. Simultaneously, the dataset extensively includes transition metals such as $Fe$, $Co$ and $Ni$, which play crucial roles in electrocatalysis, along with various other metallic elements. This ensures the model can learn diverse elemental knowledge.

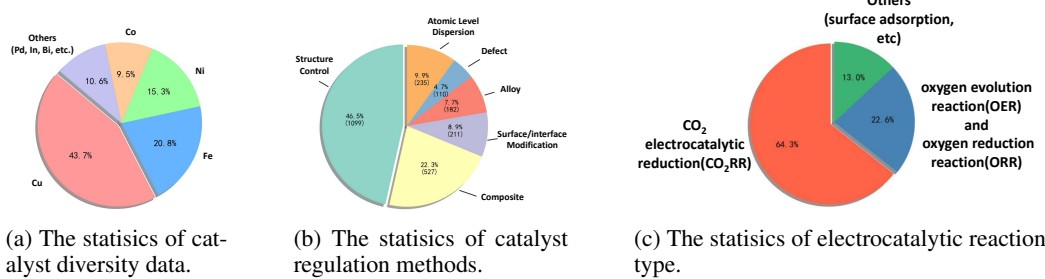

(a) The statisics of catalyst diversity data.

(b) The statisics of catalyst regulation methods.

(c) The statisics of electrocatalytic reaction type.

Figure 9: The distribution characteristics of the catalysts and reaction types involved in Catalyst-Bench.

**Distribution of Catalyst Regulation Methods:** CatalystBench incorporates a variety of advanced catalyst design and control strategies. We statistically analyze 2,364 samples from the RMC task. As shown in Fig 9b, Structure control and Composite materials are the two most prevalent approaches, accounting for 46.5% and 22.3% respectively. This reflects the current mainstream trend of enhancing catalytic performance through constructing heterojunctions and multi-component synergies. Additionally, Surface modification, Alloying, Defect engineering and Atomic-level dispersion also account for significant proportions, comprehensively covering key technologies from macroscopic morphology to atomic-scale regulation.

**Distribution of Reaction Type:** CatalystBench encompasses several core reaction types in the electrocatalysis field. As shown in Fig 9c, $CO_2$ electrocatalytic reduction($CO_2RR$) constitute the largest proportion of the dataset, accounting for 64.3%, directly attributable to our integration of multiple literature sources and datasets focused on $CO_2RR$. Simultaneously, by incorporating datasets like SACs and OC20, we have also included a substantial amount of theoretical calculation data related to oxygen evolution reaction(OER) and oxygen reduction reaction(ORR), combined as 22.6%. Additionally, a small number of other surface adsorption and catalytic reactions are included, collectively forming a diverse set of reaction scenarios.

## B    EXPERIMENTAL DETAILS

### B.1    DOMAIN-SPECIFIC CHALLENGES IN CATALYSTBENCH MULTI-TASK LEARNING

CatalystBench presents a unique configuration of tasks that diverges substantially from typical multi-task learning settings, primarily due to the heterogeneous nature of inputs, outputs and the rich domain-specific dependencies intrinsic to catalytic science. Unlike general-purpose multi-task benchmarks where tasks may share a uniform modality and output format, the catalytic design workflow necessitates the simultaneous handling of mixed input modalities comprising structured chemical descriptors, unstructured scientific text and numerical parameters. These inputs range from SMILES strings, atomic coordination tables, or facet specifications to free-form descriptions of synthesis methods and regulation schemes extracted from literature, creating a complex representational space in which feature alignment across modalities is non-trivial.

The heterogeneity further extends to output modalities. Regression tasks such as Adsorption Prediction, d-band Center Prediction and Faradaic Efficiency Prediction demand high-precision numerical values. Classification tasks, exemplified by Regulation Method Classification, require discrete categorical decisions grounded in chemical taxonomy. Generative tasks, including Material Extraction and Regulation Scheme Comprehension, produce long-form textual responses that must balance fluency and factual correctness while integrating domain knowledge. This diverse set of output spaces imposes fundamentally different optimization landscapes; mean squared error (MSE) objectives for regression tasks often operate on a much larger numerical scale than cross-entropy losses used in classification, or token-level negative log-likelihoods in generation. Without explicit control mechanisms, the disproportionately high gradients from regression losses can dominate joint training, suppressing learning in classification and generative objectives, a phenomenon that is particularly detrimental in scientific settings where each capability is equally critical.

Moreover, tasks within CatalystBench are not independent in their knowledge requirements. The Regulation Method Classification task, for instance, provides crucial mechanistic context that directly influences downstream predictive accuracy in Faradaic Efficiency estimation. Similarly, understanding textual regulation schemes is often prerequisite to correctly interpreting property trends. These upstream–downstream dependencies mean that multi-task learning in this domain must not only avoid interference between tasks, but also preserve and exploit beneficial cross-task synergies. Naively unifying outputs in a shared generative head, as in the FT setting, disregards both the modality-specific optimization needs and the asymmetric informational flow between tasks, often leading to diluted performance gains. Conversely, the MST approach, while introducing modality-specific heads, treats each task in isolation during training, forfeiting the representational transfer that arises from shared backbone exposure to the full diversity of inputs.

Empirical evidence from Fig 5 underscores these points. The FT configuration suffers notable degradation in numerical prediction tasks, reflecting loss-scale interference and insufficient modality differentiation, whereas MST exhibits limited average improvement due to the absence of multi-task semantic alignment. By contrast, the MFT strategy achieves the highest mean improvement rate of 12.44% over the ST baseline, leveraging joint backbone training to capture shared domain features, while decoupled output heads mitigate loss conflict across heterogeneous modalities. These findings substantiate that the challenges in CatalystBench, such as input heterogeneity, output space divergence, loss imbalance and domain-specific task coupling necessitate an architecture that is simultaneously modality-aware and synergy-preserving, a requirement uniquely fulfilled by the proposed MFT fine-tuning framework.

### B.2    BASELINE LLMS

The following sections will introduce the specific open-source LLMs and general closed-source LLMs we evaluate in the main text. The comparison results are presented in the main text.

**LLaMA-2** Touvron et al. (2023) is a set of large language models with parameter ranges from 7 billion to 70 billion. The model architecture remains largely unchanged from LLaMA-1, but over 40% of the data is used to train the base model. Specifically, LlaMA-2 includes pre-trained and fine-tuned models optimized for conversational applications and we have chosen to use LLaMA-2-7B as the model for comparative evaluation.

**Mistral** Jung et al. (2010) is an open-source large language model with 7 billion parameters, designed to offer high efficiency and performance on a wide range of downstream tasks. The model features a transformer architecture with various optimizations on tokenization and training data diversity. In this study, we select Mistral-7B for comparative evaluation.

**ChatGLM** GLM et al. (2024) is a bilingual large language model containing 6 billion parameters. It adopts the General Language Model (GLM) architecture and is optimized for dialogue and question-answering scenarios. ChatGLM-6B is pre-trained on extensive Chinese and English corpora, enabling strong cross-lingual generation capabilities.

**Qwen3** Yang et al. (2025) are trained on diverse multilingual datasets to improve performance across languages and domains. The model is optimized for multilingual and multitask settings, benefiting from large-scale pre-training. For evaluation purposes, Qwen3-8B is selected as a representative of mid-sized models in the Qwen3 series.

**ChemLLM** Zhang et al. (2024a) is an open-source language model deeply optimized for the chemical field, with 7B parameters. ChemLLM has converted vast amounts of structured chemical knowledge sources into over 40 million high-quality single-round Q&A pairs, known as ChemData. By fine-tuning on this large-scale instruction dataset, ChemLLM can learn a wealth of chemical facts and understand and execute diverse chemical tasks through seamless conversational interactions.

**Darwin1.5** Xie et al. (2024) is an open-source foundational LLM tailored for material science and chemistry. The model is designed with a two-stage training strategy, namely question-answering (QA) fine-tuning and multi-task learning, to enable LLM to proficiently perform chemical and materials tasks.

**Claude-3-haiku** is one member of the Claude-3 series of commercial large language models developed by Anthropic. It is designed to provide fast, safe and helpful conversational abilities, particularly for business and enterprise applications. Being a closed-source model, our interactions with Claude-3-haiku are conducted via API, employing few-shot inference to ensure fair and efficient evaluation.

**Gemini-2.5-flash-preview** Comanici et al. (2025) is a pre-release version of Google's Gemini-2.5 series, offering improvements in dialogue understanding and response speed. As a closed-source model, we access Gemini-2.5-flash-preview through its API, enabling batch evaluation for comparative studies while maintaining low overhead.

**Deepseek-v3** Liu et al. (2024a) is the third-generation model from the Deepseek series, developed to enhance code understanding, generation and natural language interactions. With advanced pre-training strategies on code and text data, Deepseek-v3 demonstrates strong capabilities across multiple tasks.

**GPT-3.5-turbo and GPT-4.** For closed-source models such as OpenAI GPT Family GPT-3.5-turbo Ye et al. (2023) and GPT-4 Achiam et al. (2023), we employ batch inference via APIs for conducting few-shot prompt inference. This approach significantly enhances evaluation efficiency and reduces overhead.

### B.3 BASELINE TRADITIONAL ML METHODS

**CatBERT** Ock et al. (2023) is a Transformer model based on the RoBERTa architecture, specifically designed for predicting adsorption energies in catalytic systems. It can replace traditional graph neural network(GNN) methods, which rely on precise three-dimensional atomic coordinates, by using human-readable text inputs that describe the "adsorbate-catalyst" system. By processing text containing key information such as adsorbate types, catalyst host material compositions and interacting atoms, CatBERTa achieves accurate predictions of catalyst adsorption energies.

**GAP-CatBERT** Ock et al. (2024) improves the accuracy of catalyst property prediction through a graph-assisted pre-training strategy. During the pre-training phase, the model utilizes a powerful graph neural network, EquiformerV2 Liao et al. (2023); Liao & Smidt (2022), to generate high-quality graph embeddings from precise atomic structures. While the CatBERTa model generates text embeddings from corresponding textual descriptions. Subsequently, this method aligns the embeddings from these two different modalities through contrastive learning, thereby transferring and injecting the rich structural knowledge contained in the graph model into the text model. The

resulting GAP-CatBERTa achieves respectable accuracy in adsorption energy prediction tasks using only text input.

**GPTchem** Jablonka et al. (2024) is an effective prediction method which combines the powerful representation capabilities of LLM with Gaussian Process Regression(GPR) strategies. In this framework, the model uses LLM fine-tuned by massive amounts of chemical text to convert discrete chemical entity information into feature embedding vectors, which are then fed into a GPR model. GPR can not only predict specific chemical properties of molecules such as formation energy, but also provide a quantified confidence interval for the prediction results through Bayesian inference.

**CGCNN** Xie & Grossman (2018) (**Crystal Graph Convolutional Neural Network**) is a graph convolutional neural network (GCN) model specifically designed for crystalline materials. Within CGCNN, the crystal structure is abstracted as a crystal graph, wherein atoms serve as the graph's nodes and chemical bonds between atoms represent the graph's edges. Node features typically represent invariant properties characterising atomic species. By stacking graph convolutional layers and pooling layers, this model learns both local and global features within the crystal structure, ultimately enabling accurate prediction of various physicochemical properties such as formation energy and bandgap.

**SchNet** Schütt et al. (2018) is a deep learning model specifically designed for molecular and periodic structures, aiming to learn potential energy surfaces and quantum chemical properties. At its core, SchNet employs continuous filtering convolutional layers to process atomic environments, with these convolutional operations being rotationally and translationally invariant. It progressively refines atomic features through interaction blocks that utilise interatomic distance information. Crucially, SchNet introduces Gaussian Radial Basis Functions (RBFs) to represent interatomic distances, enabling the model to capture long-range interactions that vary continuously with distance. The model efficiently and accurately predicts atomic energies, forces and other quantum chemical properties.

**DimeNet++** Gasteiger et al. (2020a) is a geometric deep learning model that builds upon the DimeNet Gasteiger et al. (2020b) framework, focusing on capturing angular information and tripartite interactions within molecules. Unlike traditional GNNs that solely consider atoms and bonds, DimeNet++ introduces a Directional Message Passing mechanism. It accounts not only for the connection from atom $i$ to $j$, but also for the message transmitted from atom $k$ to $i$ via $j$, thereby explicitly encoding the angular information formed by atoms $i$, $j$ and $k$. This explicit handling of the three-body term enables DimeNet++ to achieve higher accuracy and rotation invariance when predicting molecular properties, while simultaneously enhancing computational efficiency through optimised architecture.

**GemNet-OC** Gasteiger et al. (2022) is a high-performance graph neural network model, specifically designed as a variant for catalyst adsorption system architectures. Building upon DimeNet and SchNet, it further emphasises the utilisation of geometric information. By integrating explicit three-body terms and higher-order angular information, this model accurately characterises the intricate interactions between adsorbates and surface atoms on catalysts, thereby enabling high-precision predictions of key catalytic properties such as adsorption energies. It stands as one of the state-of-the-art models in the field of catalysis that relies on atomic coordinate inputs.

**Faradaic efficiency prediction** is a relatively new property prediction task in the field of catalytic materials. Gao et al. Gao et al. (2023) construct a knowledge graph of electrocatalysts based on scientific literature and propose a deep learning-based prediction model, which integrates the semantic information from the scientific literature (word embedding) with the correlation of knowledge triples (graph embedding) and realizes the prediction of the Faradaic efficiency for a targeted case.

### B.4 EVALUATION AMONG OPEN-SOURCED MODELS

Previous studies have shown that general LLMs often perform poorly when handling tasks requiring deep understanding of chemical structures, such as SMILES strings and complex chemical reasoning, as they lack internalized, domain-specific knowledge systems Bagal et al. (2021); Liu et al. (2024b); Chang et al. (2024). To select an appropriate base model, we systematically evaluated multiple existing open-source LLMs with 7B/8B parameters on representative tasks from Catalyst-Bench. The evaluation tasks encompass 3 core capabilities: structured numerical prediction, text

classification and text generation. This approach aims to comprehensively characterize the models' reasoning and generalization abilities across theoretical and experimental data conditions. The Results are shown in Table 5, which demonstrate that ChemLLM-7B maintains stable and outstanding performance across most tasks, exhibiting significant advantages over other candidate models particularly in reaction mechanism analysis and catalyst performance prediction. Therefore, we designate ChemLLM as the base model for fine-tuning to ensure subsequent experiments build upon a robust foundation of comprehensive performance.

The decision to limit the model size to small parameters stems from a balanced consideration of computational resources and application scenarios. On one hand, a 7B/8B-scale model can undergo multi-task fine-tuning on typical GPU clusters with relatively low training overhead, ensuring reproducibility and generalizability throughout the research process. On the other hand, this parameter scale already achieves satisfactory task performance in chemistry and materials domains while maintaining high operational feasibility for practical deployment.

| Model | IE | RMC | AP |
|---|---|---|---|
| **ChemLLM-7B** | **0.86** | **0.52** | **0.63** |
| Darwin1.5 | 0.84 | 0.48 | 0.59 |
| LLaMA2 | 0.76 | 0.38 | 0.24 |
| Qwen3-8B | 0.80 | 0.41 | 0.31 |

Table 5: The performance of four open-source LLMs of similar scale on representative tasks in CatalystBench.

### B.5 EVALUATION WITH TRADITIONAL ML METHODS

We compare CatalystLLM with traditional machine learning methods on 4 property prediction regression tasks in CatalystBench. The ML results for each task are obtained using specific single-task data. Taking the adsorption energy prediction task as an example, both CatBERTa and GAP-CatBERTa use the atomic text information of the catalyst, which is also the generic text in the template as Appendix D.5, as input to predict the adsorption energy value of the catalyst. These ML algorithms typically only accept one input format and do not have results for all tasks. For the AP, d-CP and FP tasks, we introduce mainstream graph neural network benchmark models currently prevalent in materials science, including CGCNN, SchNet, DimeNet++ and GemNet-OC. These models are widely regarded as standard bench-

| Model | AP | d-CP | FP |
|---|---|---|---|
| CGCNN | 0.79 | 0.66 | 0.78 |
| SchNet | 0.72 | 0.65 | 0.77 |
| DimeNet++ | 0.84 | 0.69 | 0.80 |
| GemNet-OC | 0.85 | 0.72 | 0.84 |
| CatBERTa | 0.82 | / | / |
| GAP-CatBERTa | **0.86** | / | / |
| GPTchem | / | 0.69 | **0.85** |
| CatalystLLM (ours) | 0.81 | **0.73** | 0.80 |

Table 6: The results of CatalystLLM compared with machine learning baselines for 3 prediction tasks. The best model is in bold font and the second-best is underlined.

marks in adsorption energy prediction benchmarks such as OC20. We compare CatalystLLM against publicly available checkpoints of these algorithms, all of which are trained on the same data set size as CatalystBench. It is particularly noteworthy that these graph neural network models rely on complete atomic 3D geometric coordinates and lattice parameters as inputs. They explicitly encode 3D spatial structural information through message-passing mechanisms to achieve high-precision predictions. In contrast, CatalystLLM is limited to textual descriptions extracted from experimental literature or databases, along with simplified structured text sequences such as chemical formulas, crystal plane information and adsorption site descriptions. Additionally, due to differences in input representation formats, not all tasks have ML baselines, such as text classification and semantic understanding tasks. For information extraction tasks, LLMs like the GPT series are more accurate than traditional entity extraction methods. The detail results of evaluation is shown in Table 6 and Table 7.

As can be seen from Table 6, the advantage of CatalystLLM over traditional ML methods lies in its exceptional versatility. It can provide effective predictions across a wide range of task types, whereas other models typically focus on one or two specific tasks. In terms of specific performance, CatalystLLM achieve the best score of 0.73 in the d-CP task, outperforming GPTchem. In other tasks, CatalystLLM also demonstrate strong competitiveness. For example, its adsorption energy

prediction score is 0.81 and close to SOTA model GAP-CatBERTa. In summary, while highly optimized specialized models may perform better in certain single tasks, CatalystLLM demonstrates its immense value as an efficient catalyst design LLM through its balanced and competitive performance across a wide range of catalytic prediction tasks.

Notably, when compared against graph-structured prediction models, CatalystLLM's predictive accuracy falls slightly below that of top-tier graph models like GemNet-OC and DimeNet++, which rely on precise geometric structures. Moreover, CatalystLLM outperforms earlier SchNet models. This indicates CatalystLLM's ability to effectively capture chemical knowledge and structural features implicit in textual descriptions through multitask fine-tuning. Crucially, widely-used GNNs require 3D coordinates derived from costly DFT relaxations. CatalystLLM achieves comparable accuracy using only text information.

| Model | MSE | MAE | $R^2$ |
|---|---|---|---|
| multi-layer perceptron | 0.03 | 0.15 | 0.48 |
| support vector regression | 0.06 | 0.20 | 0.11 |
| linear regression | 0.05 | 0.17 | 0.20 |
| BRR | 0.05 | 0.18 | 0.23 |
| GPR | 0.04 | 0.15 | 0.34 |
| original paper | **0.01** | **0.08** | **0.84** |
| CatalystLLM (ours) | 0.02 | 0.13 | 0.73 |

Table 7: The results of CatalystLLM compared with machine learning baselines for FEP task. The best model is in bold font and the second-best is underlined.

This enables instantaneous inference on vast chemical spaces where 3D structures are unknown. For the FEP prediction task, we introduce traditional text prediction methods from the current scientific domain. As shown in Table 7, although it falls short of specially trained state-of-the-art models for this task, CatalystLLM significantly outperforms other ML approaches, demonstrating considerable potential in this domain.

## B.6 Additional Ablation study

In addition to the ablation experiments described in the main text, we also compare the impact of different experimental settings on model capabilities to explore the crucial elements of LLM fine-tuning experiments.

### B.6.1 The impact of weighted loss-function on model performance

This experiment aims to verify the critical role of weighted loss-function calculation in multi-head Full-task fine-tuning setting. In the experimental setup described in the main text, we manually adjust the weights based on the performance of ChemLLM-MST model on the validation set for each task to balance the contribution of different tasks to the model gradient update. In addition, we use an unweighted loss-function on the Multi-head architecture, in which the weight coefficients $\lambda_{\text{task},i}$ for all tasks are set to 1.

Fig 10 shows the comparison of the results of the two sets of experiments. MFT-weighted demonstrate better performance in most tasks, proving that loss weighting is a necessary step to achieve high-performance multi-task learning. Among them, MFT-Unweighted shows a significant decline in performance in tasks such

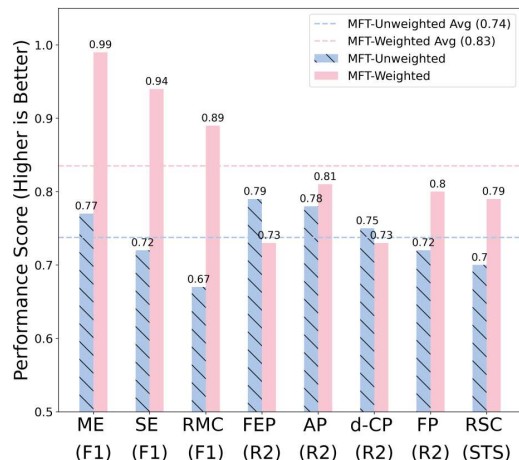

Figure 10: The impact of weighted loss-function of MFT on model performance.

as classification and information extraction. This may be because in multi-task learning, the numerical scales of the loss functions of different tasks often differ by orders of magnitude. For example, the mean squared error in regression tasks has a larger scale than the cross-entropy loss in classification tasks, leading the model to prioritize regression tasks during back-propagation. Therefore, we need to set different weight parameters to balance the contributions of different tasks to the total gradient, thereby maintaining the multi-task capability of CatalystLLM. In addition, a possible

future optimization direction is to set the weight parameters as hyperparameters and dynamically adjust them during model training.

### B.6.2 OOD LEAVE-ONE-TASK-OUT EVALUATION

To systematically evaluate CatalystLLM's generalization capabilities on unseen tasks, we design a "Leave-One-Task-Out" experimental strategy. Building upon Section 6.3 of the main text, we further conduct out-of-domain performance analysis for specific tasks. Considering the prior relationships among the 8 CatalystBench tasks—particularly the strong correlations in input modalities and knowledge dimensions for certain tasks, we select the Faradaic Efficiency Prediction (FEP) task as the holdout set. This task is excluded from model training and only evaluated during the inference phase. This task belongs to the numerical regression category, with inputs derived from structured catalyst characterization and performance data in experimental literature. It exhibits potential upstream-downstream logical connections with tasks like Regulation Method Classification (RMC) and Regulation Scheme Comprehension (RSC). FEP prediction requires comprehensive understanding and inference regarding the regulation methods, structural features and corresponding intermediate formation processes of the catalyst system.

This experiment aims to validate whether CatalystLLM can retain a certain level of performance through cross-task transfer learning and shared representation modeling, even under training conditions with complete absence of task-specific data. We compare CatalystLLM with ChemLLM-FT and representative general-purpose open-source models LLaMA2-7B and Mistral-7B under identical data partitioning and inference settings. All models are fine-tuned without FEP task training data, using the remaining seven tasks as training sets. The results are shown in Table 8.

Experimental results indicate that under the FEP-withheld setting, CatalystLLM still significantly outperforms baseline models on both $R^2$ and MAE metrics, representing an improvement of approximately 0.11 over ChemLLM-FT and exceeding 0.20 compared to general-purpose models. The outcome indicates that FEP task performance largely benefits from knowledge transfer and sharing within the multi-task architecture, particularly contributions from structural regulation tasks in terms of semantics and patterns. Furthermore, de-

| Model | $R^2$ | MAE |
|---|---|---|
| ChemLLM-FT | 0.58 | 3.21 |
| LLaMA2-7B | 0.49 | 3.87 |
| Mistral-7B | 0.51 | 3.75 |
| CatalystLLM | **0.69** | **2.84** |

Table 8: The experiments on OOD Leave-One-Task-Out evaluation on FEP task

spite performance declines compared to the original setting with FEP training data, CatalystLLM still generates predictions with physicochemical plausibility. This demonstrates that its generalization capability relies not only on task matching but also on the internal organization and expression of domain knowledge.

## C    EVALUATION DETAILS

We conduct a multi-dimensional evaluation of open-ended Q&A tasks focused on understanding and explaining catalytic regulation schemes.

### C.1    LLM-BASED EVALUATION

The RSC task aims to evaluate the model's deep understanding of catalytic regulation schemes and its text generation capabilities. The output is a segment of natural language, rather than fixed labels or numerical values. Traditional automated evaluation metrics, such as BLEU Papineni et al. (2002) or ROUGE Lin (2004) based on phrase overlap rates, cannot effectively measure the factual accuracy of answers. For example, an answer may use different words from the standard answer but still be semantically correct and conversely. Therefore, we use gpt-4o and deepseek-r1 as evaluation models M. Bran et al. (2024). They are prompted to assume the role of a catalytic field expert, but has no access to external tools such as internet browsing. We collaborate with domain experts to design a set of evaluation prompts so that LLM would imitate scientists in their thinking patterns and give scores in different dimensions: **1) Reasonableness** assesses whether the content generated by the model follows basic scientific principles and causal relationships; **2) Accuracy** assesses whether the technical terms used in the model output are consistent with recognized scientific knowledge; **3) Usability** assesses whether the answer generated by the model is practical to the question and has a certain degree of reality and operability. The LLM-based evaluation prompt we designed is shown in Appendix E.3.

### C.2    EXPERT EVALUATION

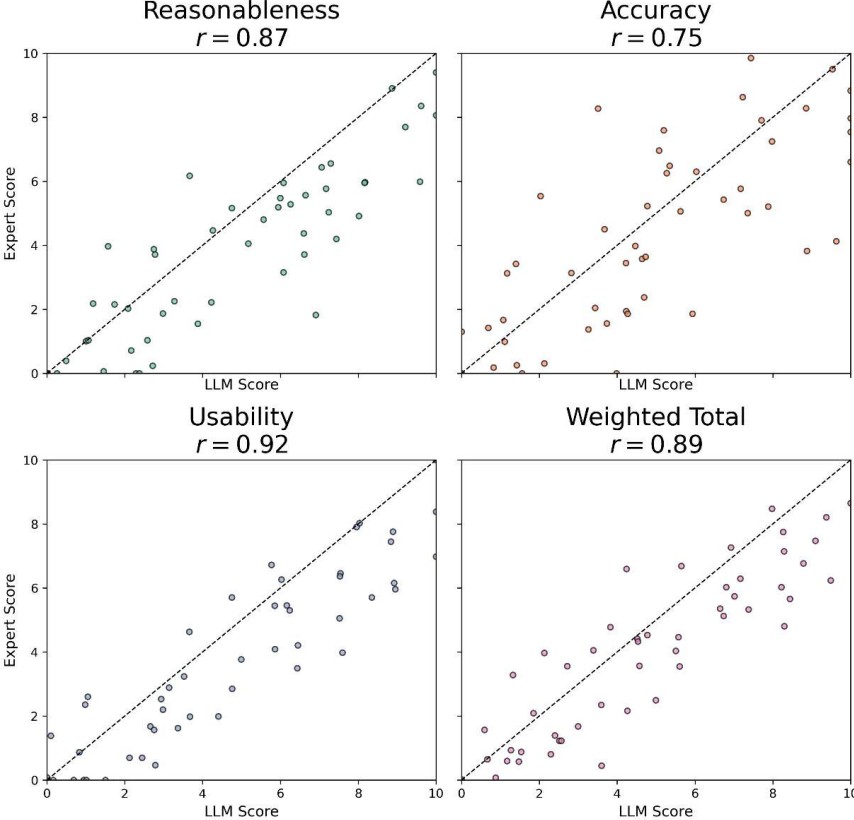

Figure 11: The 3 sub-metrics and the overall Pearson score between LLM Score and Expert Score.

Although general LLMs have vast knowledge reserves and powerful reasoning capabilities, they have a prevalent issue: the generation of hallucinations such as incorrect or fabricated information

presented as factual Xu et al. (2024). Therefore, we invite domain experts to evaluate the model's performance on a test set of 100 samples. We observe a clear trend of consistency between LLM scores and expert ratings across the overall distribution of these samples. The Pearson correlation coefficient for the average scores of the 100 samples is 0.89, indicating a high degree of linear correlation in the numerical variations between the two types of scoring. We further analyze the 3 sub-metrics and the overall score. As shown in Fig 11, samples rate highly by the LLM also tended to receive relatively high scores from experts, with a similarly pronounced consistency observed for low-scoring samples. This outcome demonstrates that despite divergences in assessing the accuracy of specific procedural details, both evaluation systems exhibit close alignment in judging overall answer quality trends. This consistency provides practical support for leveraging prompt-optimized LLM scoring during initial data screening and offers statistical justification for subsequent integration with expert review to achieve efficient quality control.

Regarding the variability in human scoring, Fig 12 shows the differences between LLM evaluation and human expert evaluation. Although human experts prefer CatalystLLM's responses based on the accuracy of catalytic materials and corresponding mechanisms, LLMs prefer GPT-4.1's answers, typically basing their evaluation on the fluency and apparent completeness of GPT-4.1 responses. Therefore, we focus on excluding GPT 4.1's answers that are complete and fluent enough to cover the question but contain incorrect information. Fig 13 shows an example of an illusion in LLM-based score and expert score. GPT4.1's answer achieves higher LLM-based score, but it makes inferences that violated the principles of catalysis, which is "K$^+$ directly participates in C-O bond cleavage". In the field of electrocatalysis, there is a non-covalent interaction mechanism whereby

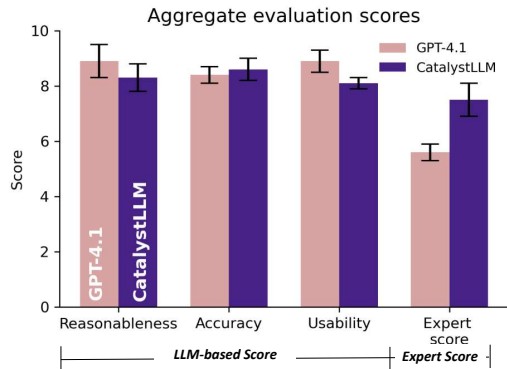

Figure 12: Average scores for gpt-4o and deepseek-r1 across three metrics and aggregated results from expert evaluators (n=100) across all tasks. Error bars indicate confidence intervals (95%).

alkali metal ions stabilize intermediates through electrostatic field effects rather than directly participating in bond formation or bond breaking. In general, LLMs tend to favor answers that are formally rigorous and logically coherent. However, in scientific evaluation, the correctness of the answer is more valuable than a complete and detailed explanation. For scientific tasks requiring real-world knowledge, LLM-based evaluation methods cannot replace expert human assessment for now.

### C.3 CASE STUDY: UNIQUE CORRECT REASONING IN COMPLEX CATALYTIC CONTEXT

To more intuitively demonstrate CatalystLLM's domain adaptation capability, we select a test case that only CatalystLLM could answer completely and correctly among all comparison models, which is shown in Appendix D.7. This task belongs to the Regulation Scheme Comprehension(RSC) category, involving a high-order coupled multi-metal catalytic system whose underlying mechanism was not directly present in the training data.

Among all comparative model responses, CatalystLLM uniquely link the inferred product shift toward methane and hydrogen to the synergistic mechanism involving *CO production at the Zn site. It accurately identifies that losing the Zn site reduces intermediate concentration and alters the reaction pathway, thereby disrupting the essential conditions for ethylene formation. This response not only aligns with established physicochemical principles but also resonates with mainstream understanding of electrocatalytic mechanisms at the mechanistic level, demonstrating the model's capacity for abstract reasoning when inferring complex sequential mechanisms. This case demonstrates that CatalystLLM can leverage cross-task learned catalytic reaction knowledge to make domain-logical inferences even without direct case training. This holds significant reference value for chemical engineers and catalytic scientists exploring novel multifunctional catalyst designs. More importantly, this case reflects the unique potential of domain-adapted LLMs in supporting high-level scientific problem-solving. Their accuracy stems not only from data coverage but also from the learned representation's ability to capture and structurally express catalytic principles during fine-tuning.

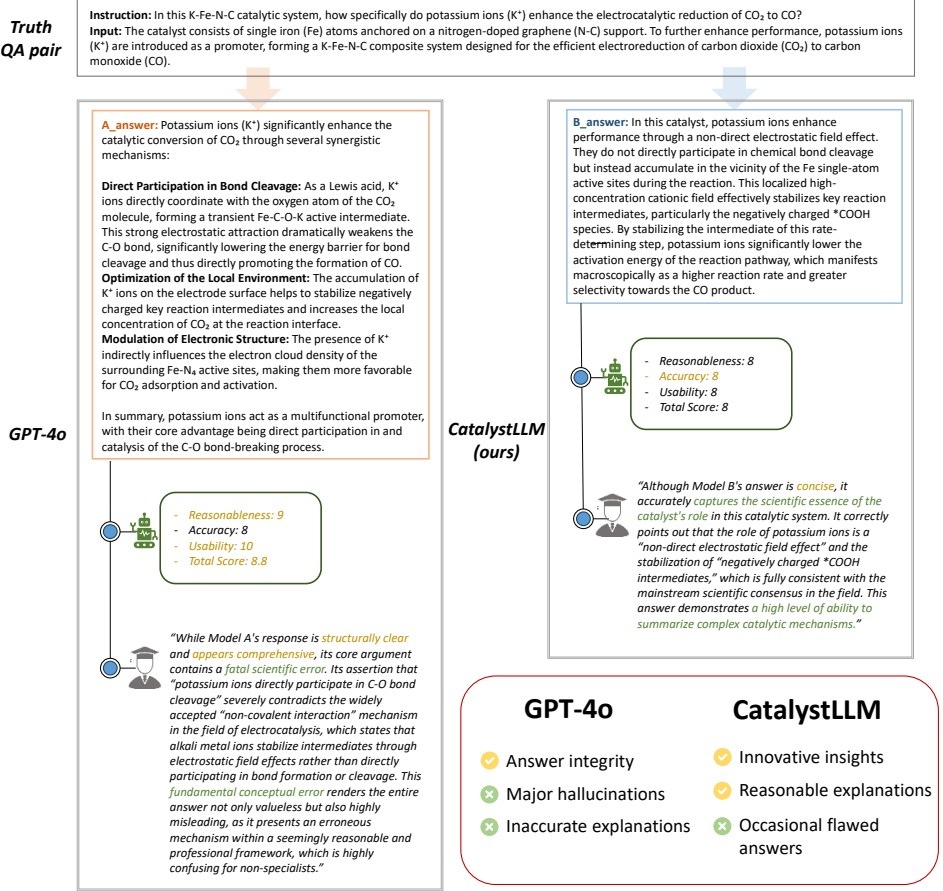

Figure 13: For the example truth Q&A pair of the same regulatory scheme semantic understanding task, the LLM-based scores and expert scores of answers given by GPT-4.1 and CatalystLLM. The yellow highlights indicate positive evaluations and the green highlights indicate negative evaluations.

## D  QUESTION TEMPLATES

In recent years, there has been a surge in the application of machine learning to chemistry, resulting in a wealth of datasets and benchmarks in chemistry and material field. However, few of these benchmarks focus on assessing LLMs for tasks specific to Catalytic Science and a standardized evaluation technique has not yet been established. This section provides concrete examples for each catalytic science task category, demonstrating how to formulate inputs and showcasing the model's expected outputs for representative problems.

### D.1  LABEL DEFINITION

<Contexts>: The text describing catalytic material control schemes or synthesis schemes in scientific literature. e.g. "The synthesized $Cu$ catalysts show relatively high crystallinity dominated by $\{111\}$ facets, promoting the generation of $CH_4$."

<Material_representation>: The actual representation of catalytic material typically shows its elemental composition or structural details, including SMILES strings, chemical formulas, CIF and other forms of existence. e.g. Cu2Pd.

<Operations>: The common operations in catalytic material synthesis schemes include ADDING, HEATING, CURING, ELECTROCHEMICAL ANODIZATION, FILTERING,

DRYING, DIPPING and REACITON. e.g. The reaction mixture was then heated at 85 °C with constant stirring for 3 h.

<Operation conditions>: The relevant attributes of catalytic material synthesis operations, including heating temperature and duration. e.g. 85 °C and 3 h

<Property>: The characteristics or properties of catalytic material describe its chemical characteristics and crucial performance, including adsorption energy, d-band center, formation energy and actual catalytic Faradaic efficiency. e.g. 90%.

## D.2   MATERIAL EXTRACTION TEMPLATES

---

**Material Extraction Templates**

**Instructions:**

- **What specific materials are mentioned as catalysts in the text?**

- **List all keywords that refer to catalyst materials or their types discussed in the article.**

- **Which terms in the document are associated with or describe catalyst materials?**

**Input:** <Contexts>

**Output:** <Material_representation>

---

## D.3 SYNTHESIS EXTRACTION TEMPLATES

---

**Synthesis Extraction Templates**

**Instructions:**

- **Round 1: What is the target material synthesized in the given text?**

**Input:** <Contexts>

**Output:** The target material synthesized in the text is <Material_representation>.

- **Round 2: List the main experimental operations described in the synthesis process.**

**Input:** <Contexts>

**Output:** The experimental operations mentioned are <Operations>

- **Round 3: For <Operations>, specify the relevant <Operation conditions>.**

**Input:** The reaction mixture was then heated at 85 °C with constant stirring for 3 h.

**Output:** The <Operation conditions> is 800°C.
...

---

## D.4 REGULATION METHOD CLASSIFICATION TEMPLATES

---

**Regulation Method Classification Templates**

**Instructions:**

- **What is the type of regulation method applied to the catalytic material in the described scheme?**

- **Which regulation approach or technique is used to modify the catalytic material as detailed in the text?**

- **Identify the specific method of regulation implemented in this catalytic material scheme.**

**The options:** A. Alloy B. Defect C. Atomic level dispersion D. Surface/interface modification E. Composite F. Structure control

**Input:** Cu/Bi bi-metal compounds derived from MOFs.

**Output:** E. Composite.

---

## D.5 PROPERTY PREDICTION TEMPLATES-GENERAL TEXT

---

**Property Prediction Templates**

**Instructions:**

- **What is the $<$Property$>$ of the catalytic material represented by this $<$Material_representation$>$?**

- **Can you provide the $<$Property$>$ associated with this catalytic material $<$Material_representation$>$ for reduction of CO2 to produce CO?**

- **What is the $<$Property$>$ for the catalytic material described by $<$Material_representation$>$?**

**Input:** $CHTi_{18}Pd_{54}$ (2 1 0) [C Ti Pd Pd hollow [Ti Pd Pd Pd Pd Pd Pd ...]].

**Output:** -1.6033 eV.

---

## D.6 PROPERTY PREDICTION TEMPLATES-CHEMICAL TEXT

---

**Property Prediction Templates**

**Instructions:**

- **Calculate the adsorption energy of CH species on the $Ti_18Pd_54$(210) surface using the $<$Material_representation$>$:**

**input:**

- **Adsorbate:** CH
- **Catalyst:** $Ti_18Pd_54$
- **Facet:** (2 1 0)
- **Active site:** hollow site coordinated by C, Ti, Pd, Pd

**Output:** -1.6033 eV.

---

## D.7 REGULATION SCHEME COMPREHENSION

---

**User**

**Instruction:**
How does the core-shell architecture of CoP@NiFe-LDH specifically address the common limitations of using NiFe-LDH alone for the Oxygen Evolution Reaction (OER)?

**Method Description:**
To construct a highly efficient catalyst for the Oxygen Evolution Reaction (OER), a hierarchical core-shell structure was designed. The core consists of conductive cobalt phosphide (CoP) nanowires, which serve as a scaffold. Vertically aligned nickel-iron layered double hydroxide (NiFe-LDH) nanosheets were then grown directly on the surface of the CoP nanowires. The resulting composite material, denoted as CoP@NiFe-LDH, leverages the high intrinsic activity of the NiFe-LDH shell and the metallic properties of the CoP core.

---

**CatalystLLM**

[Answer]
The CoP@NiFe-LDH core-shell architecture strategically overcomes two primary limitations of using standalone NiFe-LDH as an OER catalyst: its poor intrinsic electrical conductivity and the tendency for nanosheets to aggregate, which reduces the number of available active sites.

**User**

**Instruction**:
Based on the described tandem catalysis mechanism, if the zinc (Zn) sites in this catalyst were to be completely removed or deactivated, leaving only the functional copper (Cu) sites, what change is expected for the catalyst's major product? Please explain the reasoning.

**Method Description:**
To efficiently convert carbon dioxide ($CO_2$) into high-value ethylene ($C_2H_4$), researchers designed a dual-function catalyst. This catalyst consists of atomically adjacent copper (Cu) and zinc (Zn) sites dispersed on a nitrogen-doped carbon (NC) support, denoted as Zn-Cu/NC. The core mechanism is a "tandem catalysis" process: first, the Zn sites act as efficient "CO factories," selectively reducing $CO_2$ to the carbon monoxide (*CO) intermediate. Subsequently, due to spatial proximity, the generated *CO intermediate rapidly migrates to the adjacent Cu sites, where the critical C-C coupling reaction occurs to ultimately produce ethylene ($C_2H_4$).

**CatalystLLM**

[Answer]
Removing the zinc (Zn) sites would shift the catalyst's selectivity from ethylene ($C_2H_4$) to methane ($CH_4$) and hydrogen ($H_2$). This is because eliminating the Zn "*CO factory" breaks the synergistic tandem mechanism, depriving the copper (Cu) sites of the high concentration of *CO intermediates needed for C-C coupling. Consequently, the catalyst would behave as a less efficient, single-function copper system, where the reaction favors deep reduction to methane or hydrogen evolution.

# E  PROMPTS

Here are the prompt instructions we build during the benchmark construction and evaluation process.

## E.1  PROMPTS FOR QA GENERATING

> **<Q&A Generation>Prompt**
>
> **Please complete the following tasks based on the text of the catalytic regulation scheme provided below and its context:**
> **1. Task requirements:**
> **- Extract core keywords related to the catalytic scheme [Method description] from the text and context of the regulation scheme (e.g., reaction type, material system, innovative mechanisms, performance enhancement points, etc.).**
> **- Based on the extracted keywords, identify 3–5 key questions regarding the characteristics, advantages and improvements of this regulation scheme compared to traditional catalytic schemes and provide answers based on the scheme and its context.**
> **- Questions should focus on: regulation schemes, material innovation, performance advantages and practical application prospects. Avoid asking simple or definitional questions.**
> **- Assume that the user does not have access to the original paper or any external sources, so ensure that the questions and answers are self-contained. - Answers should be concise and specific, based on textual and contextual [Contexts] facts.**
> **2. Example:**
> **Input:**
> **- Method description:{example_regulation_method_description}**
> **- Contexts: {example_contexts}**
> **3. output format:**
> **- Please present the generated question-answer pairs in the following format:**
> **Q1: [Question 1]**
> **A1: [Answer 1]**
> **...**
> **Q5: [Question 5]**
> **A5: [Answer 5]**
> **- Use of multiple sentence structures. Questions need to be phrased in a way that is easy to understand.**
> **Input:**
> **- Method description: {regulation_method_description}**
> **- Contexts: {contexts}**

## E.2  PROMPTS FOR CHEMICAL-TEXT GENERATION

In cheminformatics, SMILES (Simplified Molecular Input Line Entry System) is a widely used method for representing molecules as linear strings. It encodes a molecule's topological structure, atom types, bond types and partial stereochemical information using a sequence of ASCII characters. This compact format facilitates storage and transmission. However, this encoding relies on a character sequence governed by syntactic rules, with structural information implicit within the string pattern. For language models lacking specialized chemical parsing components, extracting relationships between atoms and bonds requires symbolic parsing, resulting in relatively low information explicitness.

| Task Type | General Text | Formatted Text |
|---|---|---|
| RMC | "Nano-porous $Au_3Cu$ alloy" | "Matrial: $Cu$, Scheme: $Au_3Cu$, Structure: Nano-porous" |
| FEP | "Nano-porous $Au_3Cu$ alloy material catalyzes $CO_2$ reduction to produce $CO$" | "Material: $Cu$, Scheme: $Au_3Cu$, Structure: Nano-porous, Reactant: $CO_2$, Product: $CO$" |
| AP | "$FeN4-graphene*OH$" | "Active_center: $Fe$, Coordination: $N_4$, Support: graphene, Adsorbate: $*OH$" |

Table 9: The examples of different input format for CatalystBench tasks.

In the comparative experiments of Section 6.3, we employ not only plain SMILES strings but also introduced structured molecular representations, such as explicit bond lists or atomic property tables. Table 9 shows information on two input strategies from the same original catalyst dataset. These representations are theoretically equivalent to SMILES in information content, covering all connectivity information of the molecular structure. However, the structured format directly presents atomic indices, bond types and connectivity relationships, encoding the molecular graph topology as explicit data structures in the input. Since this representation aligns with LLMs' parsing patterns for tabular and graph-structured data, it reduces the model's need to infer relationships from character sequences, making it easier for the model to correctly understand and utilize the information.

---

**<Catalytic chemical-text generation>Prompt**

**You are an expert in materials science and chemistry. Your goal is to parse the unstructured, natural language text describing a catalytic material and convert it into a structured JSON object.**
**1. Task requirements:**
**- Analyze the [Material general text]: Carefully read the description of the catalytic material.**
**- Identify Components: Extract the distinct chemical and structural components of the catalyst system, including:**
**The primary catalytically Active Component (the core material driving the reaction).**
**The Support or Substrate on which the active component is loaded.**
**Any Dopants, Promoters, or Modifiers introduced to alter the properties.**
**- Extract Key Features: Identify crucial characteristics for each component, such as material composition, morphology (e.g., nanoparticle, nanosheet, single-atom), crystal facets, size and specific properties (e.g., conductive, porous).**
**- Generate Structured Formula: Create a concise, standardized formula-like string that represents the overall catalyst architecture (e.g., Active@Support, Dopant-Active/Support).**
**- Format as JSON: Organize all extracted information into a clean, hierarchical JSON object according to the specified Output Format. If a category is not mentioned in the text, use an empty list [].**
**2. Example:**
**Input:**
**- Material general text: {example_material_general_text}**
**3. output format:**
**- Material chemicl text: {material_json}.**
**- The output must be a single, valid JSON object with the following structure. Do not include any explanatory text outside of the JSON block.**
**Input:**
**- Material general text: {material_general_text}**

## E.3 PROMPTS FOR LLM-BASED EVALUATION

> **<LLM-based evaluation>Prompt**
>
> **Please complete the following tasks based on the truth QA pair and two answers by model A and model B provided below and its context:**
> **1. Task requirements:**
> **- Extract core keywords related to the context of the regulation scheme from [Truth answer](e.g., reaction type, material system, innovative mechanisms, performance enhancement points, etc.).**
> **- Extract the core content of the [Answer by model A] and [Answer by model B] without changing the original content.**
> **- Based on the extracted keywords, Score the core content of the two answers according to the following scoring criteria. Provide specific details and explanations for any deductions.**
> **2. Scoring criteria:**
> **- Reasonableness: whether the answer is consistent with the description of the regulation method in the question. If the answer is consistent, 10 points will be given and if it is not, 0 points will be given.**
> **- Accuracy: whether the answer is consistent with the key elements of the correct answer. 3 points if one key element is included, 6 points if two key elements are included and 10 points if all are consistent.**
> **- Usability: whether the answer actually answers the question. If the response in unreliable which means it does not have any correct facts, 0 points will be given, If the response is reliable in its own right but does not match the question, 5 points will be awarded and if the response actually answers the question, 10 points will be awarded.**
> **2. Example:**
> **Input:**
> **- Truth question: {example_truth_question}**
> **- Truth answer: {example_truth_answer}**
> **- Answer by model A: {example_A_answer}**
> **- Answer by model B: {example_B_answer}**
> **3. output format:**
> **- Please rate the [Answer by model A] and [Answer by model B] out of 10 in terms of reasonableness, accuracy and usability. Reasonableness is 20%, accuracy is 50% and usability is 30%.**
> **Reasonableness: [Score_reasonableness]**
> **Explanation for reasonableness: [Explanation_reasonableness]**
> **Accuracy: [Score_accuracy]**
> **Explanation for accuracy: [Explanation_accuracy]**
> **Usability: [Score_usability]**
> **Explanation for usability: [Explanation_usability]**
> **Total score: [Total_score]**
> **Analysis: [Analysis]**

## F STATEMENT ON THE USE OF LARGE LANGUAGE MODELS

During the preparation of this manuscript, large language models (LLMs) were employed in a limited capacity to assist with linguistic refinement, grammar correction and improvement of clarity in the English prose. The scientific content, experimental design, data analysis and interpretation of results are entirely the work of the authors and no LLM was used for the generation of novel scientific claims or data fabrication. All factual statements, numerical results and interpretations have been manually verified by the authors to ensure accuracy and integrity. The assistance of LLMs was restricted to improving readability and cohesion and its use complies with prevailing academic publishing policies concerning AI-assisted writing.

