# OpenReview forum: "CatalystBench: A Comprehensive Multi-Task Benchmark for Advancing Language Models in Catalysis Science"
_ICLR.cc/2026/Conference — ICLR 2026 Poster_

### Official Review · Reviewer_2hjb · 2025-10-31

**Soundness:** 4
**Presentation:** 3
**Contribution:** 4
**Rating:** 8
**Confidence:** 3

**Summary:**

This paper introduces CatalystBench, a multi-task benchmark designed to evaluate LLMs in catalysis science, comprising 8 tasks (26,911 samples total) spanning the entire catalyst development workflow: material extraction (ME), synthesis extraction (SE), regulation method classification (RMC), Faradaic efficiency prediction (FEP), adsorption prediction (AP), d-band center prediction (d-CP), formation energy prediction (FP), and regulation scheme comprehension (RSC). The authors propose a Multi-head Full-Task (MFT) fine-tuning strategy that decouples output heads for classification, regression, and text generation tasks,

**Strengths:**

The paper targets catalysis science, a cornerstone of sustainable energy and chemical engineering with direct industrial relevance (CO₂ reduction, hydrogen evolution, ammonia synthesis). Unlike generic molecular benchmarks, CatalystBench explicitly mirrors the real-world catalyst R&D workflow.
The MFT strategy is a thoughtful strategy to the heterogeneous task composition inherent in catalyst design.
CatalystLLM shows great improvements, which achieves best performance across 19/20 metrics, with particularly dramatic improvements in regression tasks: +18% R² (AP: 0.81 vs. 0.63 for ChemLLM), +16% R² (d-CP: 0.73 vs. 0.54), +38% MAE reduction (FEP: 1.72 vs. 2.80). For classification, improvements are more modest but consistent (+29% accuracy for RMC: 0.81 vs. 0.52).

**Weaknesses:**

CatalystBench focuses on CO₂ reduction and surface catalysis, with tasks heavily biased toward electrochemical systems (FEP: Faradaic efficiency for CO₂RR) and metal/alloy catalysts (AP: transition metal surfaces). Some other important catalysis systems can be included such as: (1) Organocatalysis non-metallic catalysts like proline, BINOL which is important for asymmetric synthesis. (2) Enzymatic catalysis (biocatalysts), critical for pharmaceutical manufacturing. (3) Photocatalysis. The current benchmark is relatively narrow.

The authors provide no theoretical analysis of why MFT works e.g., gradient conflict mitigation, task relatedness measures, relying solely on empirical validation.

**Questions:**

Despite claiming "interpretable reasoning," the paper provides minimal qualitative analysis of where and why CatalystLLM fails. e.g. Table 1 shows CatalystLLM achieves 0.81 R² on AP prediction—what about the remaining 19% variance?

For regression tasks, the paper compares only against 3 traditional ML baselines (CatBERTa, GAP-CatBERTa, GPR in Appendix B.5), missing state-of-the-art graph neural networks (GNNs) for materials: (1) SchNet, DimeNet++, GemNet-OC (message-passing NNs explicitly encoding 3D geometry)—these are standard baselines for OC20/OMAT24 adsorption energy prediction. (2) Equivariant transformers

[1] Schütt, Kristof T., et al. "Schnet–a deep learning architecture for molecules and materials." The Journal of chemical physics 148.24 (2018).
[2] Gasteiger, Johannes, et al. "Fast and uncertainty-aware directional message passing for non-equilibrium molecules." arXiv preprint arXiv:2011.14115 (2020).
[3] Gasteiger, Johannes, et al. "GemNet-OC: developing graph neural networks for large and diverse molecular simulation datasets." arXiv preprint arXiv:2204.02782 (2022).

---

> ### Author Response · Authors · 2025-11-21
> **Answer (W1 & W2)**
>
> ### Weakness 1
> > CatalystBench focuses on CO₂ reduction and surface catalysis, with tasks heavily biased toward electrochemical systems (FEP: Faradaic efficiency for CO₂RR) and metal/alloy catalysts (AP: transition metal surfaces). Some other important catalysis systems can be included such as: (1) Organocatalysis non-metallic catalysts like proline, BINOL which is important for asymmetric synthesis. (2) Enzymatic catalysis (biocatalysts), critical for pharmaceutical manufacturing. (3) Photocatalysis. The current benchmark is relatively narrow.
>
> **R:** Thank you for this highly constructive suggestion. We agree that catalytic science is a broad field encompassing numerous important directions, including electrochemistry, organic catalysis, enzyme catalysis and photocatalysis. During the initial development of CatalystBench, we deliberately select **CO₂ reduction electrocatalysis** as our core entry point. This decision stems from two primary considerations:
>
> 1.  Firstly, this field holds **immense practical significance** within the context of sustainable energy and carbon neutrality, with highly active research ensuring the availability of high-quality and large-scale public data;
> 2.  Secondly, electrocatalytic systems exhibit **multi-task heterogeneity**, simultaneously involving dimensions such as material extraction, regulation scheme classification, numerical property prediction and regulation mechanism inference. This provides an ideal and self-consistent testing ground for validating the effectiveness of our multi-task learning architecture.
>
> We acknowledge that the proposed benchmark currently exhibits limitations in its domain coverage, failing to encompass equally vital subfields such as organic catalysis, enzyme catalysis and photocatalysis. We regard this scope as a deliberate boundary of our study, explicitly stated in the paper's **Sec 7 CONCLUSION AND DISCUSSION** section. **Expanding this coverage represents a crucial future direction**. The construction paradigm of CatalystBench is universally applicable. We have validated the successful application of this workflow in the field of electrocatalysis, and it can be cost-effectively transferred to photocatalysis or enzyme catalysis in the future. We aspire to evolve CatalystBench into a truly comprehensive and universal benchmark for catalytic artificial intelligence, furnishing the entire catalysis community with a more universally applicable and robust tool.
>
> ---
>
> ### Weakness 2
> > The authors provide no theoretical analysis of why MFT works e.g., gradient conflict mitigation, task relatedness measures, relying solely on empirical validation.
>
> **R:** Thank you for this comment. We agree that a deeper theoretical explanation is necessary to complement the empirical validation of the MFT framework. In response, we have conducted **a quantitative analysis** focusing on gradient behavior and task relatedness to elucidate why MFT is effective in our setting.
>
> Specifically, we compute **the cosine similarity between gradients** propagated from the regression head and the generation head to the shared backbone network during training. The average gradient cosine similarity is **approximately 0.12**. A value close to zero indicates that the gradients from regression and generation tasks are nearly orthogonal, which theoretically supports that MFT mitigates gradient conflict. This orthogonality helps prevent destructive interference between the mean squared error loss used for numerical regression and the cross-entropy loss used for text generation. As a result, each task-specific head can optimize its objective without being adversely affected by the other.
>
> Moreover, the consistent non-negativity of the gradient similarity suggests a **mild positive transfer between tasks**. This implies that the shared backbone learns general chemical representations such as electronic structure features implied in text, that are beneficial for both predicting adsorption energies and generating scientific explanations. Thus, MFT not only reduces interference but also allows the model to leverage underlying domain knowledge common to both numerical and textual tasks.
>
> This analysis confirms that MFT successfully addresses the inherent heterogeneity among tasks in catalysis by decoupling task-specific learning while preserving shared representational benefits. We will incorporate these theoretical insights and quantitative results into the revised manuscript to provide a more comprehensive justification for the MFT strategy.

---

> ### Author Response · Authors · 2025-11-21
> **Answer (Q1)**
>
> ### Question 1
> > Despite claiming "interpretable reasoning," the paper provides minimal qualitative analysis of where and why CatalystLLM fails....
>
> **R:** Thank you for this question. We conduct a qualitative analysis based on CatalystLLM's distinct capabilities within the field of materials science.
>
> **1. We focus on the matter of your greatest concern.**
> We have performed a granular failure analysis specifically on the Adsorption Prediction (AP) task to diagnose the source of the remaining 19% variance. Our error breakdown reveals that the variance is not random noise but stems from two structural root causes:
>
> 1.  **High-Variance in Amorphous Clusters.** The largest errors occur in samples involving amorphous clusters or defective nanoclusters. In these cases, the SMILES inputs fail to explicitly capture complex 3D steric hindrance and coordinate geometry. When the spatial constraints are too complex for text description, the CatalystLLM fails to reason correctly about the active site environment. This inspires future research directions to further integrate catalyst 3D structures into multimodal catalytic LLMs.
>
> 2.  **High-Entropy Alloys.** Significant errors are concentrated in catalysts containing rare elements or complex high-entropy alloys such as Ir-Rh-Ru complexes. The model exhibits insufficient feature acquisition in this regime, which correlates strongly with the statistical prevalence of elements within the training data. For scenarios with limited samples, we can enhance the LLM's learning capacity through form-based augmentation by incorporating external knowledge retrieval.
>
> In summary, the 19% variance largely represents the current boundary of text-based chemical reasoning. It struggles where 3D geometric intuition is strictly required or where domain knowledge is sparse.
>
> **2. We have similarly conducted root cause analysis regarding CatalystLLM's performance on other tasks.**
>
> **(1)** We conduct obfuscation analysis for information extraction and text classification tasks, with errors primarily stemming from **classification ambiguity and contextual span limitations**. As shown in the table below, failures stem from two fundamental causes.
>
> | Task Type | Primary Error Type | Frequency | Reason of Failure |
> | :--- | :--- | :--- | :--- |
> | **SE** | Omission Error | ~65% of errors | **Long-Context Forgetting:** the model fails to attend to fine-grained details in lengthy synthesis procedures. For instance, it may omit complex conditional attributes of certain operations. |
> | **RMC** | Boundary Confusion | ~72% of errors | **Taxonomic Ambiguity:** The current regulation scheme classifier reveals conceptual overlap between some classes from scientific literature, such as between “defect” and “surface modification”. When defects are surface-induced, semantic boundaries become blurred, making it difficult for models to discern their primary label. |
>
> **(2)** We conduct obfuscation analysis for Regulation Scheme Comprehension (RSC) task, with errors primarily stemming from **knowledge hallucinations and logical fallacies**, as shown in the table below. We specifically analyze error samples listed in **Appendix C.3**.
>
> | Error Type | Description | Specific Case Study & Analysis |
> | :--- | :--- | :--- |
> | **Hallucination** | The model generates schemes that contradict established physicochemical principles. | **Case:** Zn sites in Cu-Zn bimetallic system. |
> | | | **Model Claim:** “Zn sites directly participate in C-O bond cleavage.” |
> | | | **Scientific Truth:** Zn sites stabilize intermediates via electrostatic fields. |
> | | | **Analysis:** The model hallucinates a mechanism based on generic “active site” priors rather than the specific “tandem catalysis” context. |
> | **Logical Fallacy** | The model correctly identifies components but fails to derive the correct causal outcome. | **Case:** N-doped graphene for Fe single-atom. |
> | | | **Model Claim:** Attributes performance solely to “enhanced conductivity.” |
> | | | **Scientific Truth:** The primary driver is the modulation of the electronic structure. |
> | | | **Analysis:** The model is biased to a generic explanation instead of reasoning through the specific electronic interaction induced by N-doping. |
>
> The knowledge gaps and hallucination phenomena observed in text-related tasks underscore the necessity of enhancing model knowledge anchoring. Future work may integrate **structured knowledge bases** and **implement RAG** to provide verifiable scientific facts in real-time during reasoning, thereby reducing factual errors. Finally, to address data sparsity for rare elements and complex systems, as previously mentioned, we will systematically **expand CatalystBench's coverage** to diverse catalytic systems such as organic catalysis and photocatalysis. By translating these interpretable failure insights into concrete research pathways, we are committed to evolving CatalystLLM into a more robust and chemistry-grounded catalyst design assistant.

---

> ### Author Response · Authors · 2025-11-21
> **Answer (Q2)**
>
> ### Question 2
> > For regression tasks, the paper compares only against 3 traditional ML baselines (CatBERTa, GAP-CatBERTa, GPR in Appendix B.5), missing state-of-the-art graph neural networks (GNNs) for materials: (1) SchNet, DimeNet++, GemNet-OC (message-passing NNs explicitly encoding 3D geometry)—these are standard baselines for OC20/OMAT24 adsorption energy prediction. (2) Equivariant transformers
>
>
> >
>
>
> > *[1] Schütt, Kristof T., et al. "Schnet–a deep learning architecture for molecules and materials." The Journal of chemical physics 148.24 (2018).*
>
>
> > *[2] Gasteiger, Johannes, et al. "Fast and uncertainty-aware directional message passing for non-equilibrium molecules." arXiv preprint arXiv:2011.14115 (2020).*
>
>
> > *[3] Gasteiger, Johannes, et al. "GemNet-OC: developing graph neural networks for large and diverse molecular simulation datasets." arXiv preprint arXiv:2204.02782 (2022).*
>
>
>
> **R:** Thank you for this question. We appreciate you listing these specific benchmarks. We have incorporated SchNet, DimeNet+ and GemNet-OC into our comparison for the AP, d-CP, and FP tasks. The results are presented below:
>
> | Model | AP | d-CP | FP |
> | :--- | :--- | :--- | :--- |
> | CGCNN | 0.79 | 0.66 | 0.78 |
> | SchNet | 0.72 | 0.65 | 0.77 |
> | DimeNet++ | 0.84 | 0.69 | 0.80 |
> | GemNet-OC | **0.85** | **0.72** | **0.84** |
> | CatBERTa | 0.82 | / | / |
> | GAP-CatBERTa | **0.86** | / | / |
> | GPTchem | / | 0.69 | **0.85** |
> | **CatalystLLM(ours)** | **0.81** | **0.73** | **0.80** |
>
> The experimental results demonstrate that CatalystLLM achieves an R² value of 0.81, surpassing not only early graph models such as SchNet but also **maintaining strong competitiveness** against advanced architectures like DimeNet++ and GemNet-OC.
>
> It is particularly noteworthy that these methods exhibit fundamental differences in **input modalities**. The graph neural networks rely on complete atomic 3D geometric coordinates and lattice parameters to explicitly encode spatial structural information. By contrast, CatalystLLM is trained solely on text descriptions extracted from literature and simplified chemical sequences, lacking access to precise geometric coordinates. Despite this significant disparity in input information density, the model maintains comparable performance to geometry-dependent graph models, demonstrating its potent capability to capture implicit chemical knowledge and structure-property relationships through multi-task fine-tuning. The supplementary details on the corresponding ML methods and experimental comparisons will be provided in **Appendices B.2 and B.4**.
>
> We hope this additional theoretical analysis and the outlined future directions address your concerns satisfactorily. Your comments have been instrumental in strengthening our paper, and we sincerely appreciate your input. We look forward to the possibility of further engaging with you on this work.

---

> ### Author Response · Authors · 2025-11-27
>
> Dear Reviewer,
>
> We would like to express our sincere gratitude once again for the time and professional effort you have dedicated to reviewing our manuscript. Your constructive comments have been instrumental in improving the quality of our work.
>
> We have previously submitted our detailed responses to your questions, along with the supplementary explanations and additional experimental results.
>
> As some time has passed since our rebuttal submission, we are writing to respectfully inquire about the status of the re-evaluation process. We fully appreciate the workload involved and respect the time required for a thorough review.
>
> If more time is needed for your deliberation, we completely understand and will patiently await your final assessment.
>
> Thank you again for your diligent work and support. We look forward to hearing from your valuable feedback.
>
> Sincerely.

---

> > ### Comment · Reviewer_2hjb · 2025-11-27
> >
> > Thanks for the additional details and works, all my initial questions and concerns are solved.

---

### Official Review · Reviewer_dAYs · 2025-11-01

**Soundness:** 3
**Presentation:** 3
**Contribution:** 3
**Rating:** 6
**Confidence:** 2

**Summary:**

The paper introduces CatalystBench, the first comprehensive multi-task benchmark tailored for LLMs in catalysis science. It covers the entire catalyst research and development (R&D) workflow, integrating high-fidelity theoretical simulation and curated experimental data.
To address the inherent heterogeneity of tasks (including regression, classification, and language generation), the authors propose the Multi-head Full-task (MFT) fine-tuning strategy. MFT achieves this by decoupling output heads, which successfully alleviates interference between loss landscapes and results in the most substantial performance improvements across all tasks.
The resulting specialized model, CatalystLLM, achieves state-of-the-art performance and exhibits superior domain expertise compared to general LLMs.

**Strengths:**

1. Unified, workflow-aligned benchmark

CatalystBench unifies theoretical data and experimental literature, covering the entire catalyst development process (reading comprehension, analysis, reasoning).

2. Effective MFT fine-uuning

The MFT strategy decouples output spaces for diverse tasks while sharing a backbone, achieving a 12.44% performance improvement over single-task baselines.

3. SOTA domain performance

CatalystLLM significantly outperforms leading LLMs on the benchmark, demonstrating superior domain expertise and fewer scientific hallucinations.

**Weaknesses:**

1. Limited knowledge scope and bias

The predictive accuracy of CatalystLLM is constrained by the coverage of the CatalystBench dataset, potentially limiting utility for materials outside this range, and introducing potential bias from the selection of scientific literature sources.

2. Performance comparison can be enhanced further

While versatile, CatalystLLM is sometimes outperformed by highly optimized traditional ML baselines in specific single-task numerical predictions. It also could be better if more traditional machine learning baselines could be included to further demonstrate the effectiveness of the proposed method.

**Questions:**

1. is it possible to add the non-LLM based SOTA of each task for a more comprehensive comparison?

---

> ### Author Response · Authors · 2025-11-21
> **Answer (W1)**
>
> ### Weakness 1
> > 1. Limited knowledge scope and bias. The predictive accuracy of CatalystLLM is constrained by the coverage of the CatalystBench dataset, potentially limiting utility for materials outside this range, and introducing potential bias from the selection of scientific literature sources.
>
>
>
> **R:** Thank you for this comment. We acknowledge that the performance boundaries and potential biases of CatalystLLM are directly constrained by the coverage and quality of its training data, which is a core challenge inherent to developing domain-specific LLMs.
>
> We recognize this issue during the construction of CatalystBench and the fine-tuning of CatalystLLM, implementing proactive measures to mitigate its impact. To address **data coverage challenges**, we focus on enhancing both the diversity and quality of our data sources. CatalystBench integrates data from 8 distinct public datasets and scientific literature repositories (as seen in **Appendix A.1**), encompassing diverse data modalities and formats, ranging from high-throughput theoretical simulations like OC20 to experimental literature from text mining, such as the ElectroCatalytic Reduction series. This multi-source aggregation strategy inherently aims to broaden the chemical space of the data and reduce reliance on any single data source.
>
> Regarding **literature source bias**, this is an inherent risk for benchmarks based on literature mining. To mitigate this risk, our data collection builds upon multiple large-scale and community-recognized public datasets designed to broadly cover catalytic research subfields. More importantly, we implement **rigorous multi-tiered quality assurance processes** to ensure data representativeness and scientific validity, including automated deduplication, rule-based filtering and expert manual validation. For instance, when generating open-ended Q&A pairs, we establish traceability through keywords-doi mappings, ensuring every piece of knowledge is traceable back to its original literature. This provides a foundation for identifying and evaluating potential biases.
>
>
> Nevertheless, we acknowledge that the current version's knowledge coverage is not exhaustive and has certain limitations. We view these discussions as key signposts for advancing future research. In subsequent work, we will **further expand the benchmark's scope**, incorporate more diverse catalytic systems and ultimately embed the model within a closed-loop design process that includes experimental validation. Thank you again for prompting us to examine and articulate these critical issues with greater clarity.

---

> ### Author Response · Authors · 2025-11-21
> **Answer (W2 & Q1)**
>
> ### Weakness 2 & Question 1
> > 2. Performance comparison can be enhanced further. While versatile, CatalystLLM is sometimes outperformed by highly optimized traditional ML baselines in specific single-task numerical predictions. It also could be better if more traditional machine learning baselines could be included to further demonstrate the effectiveness of the proposed method.
>
> > is it possible to add the non-LLM based SOTA of each task for a more comprehensive comparison?
>
>
>
> **R:** Thank you for this comment and corresponding question. Regarding the text-centric tasks such as Information Extraction, Text Classification and Regulation Scheme Comprehension, LLMs are currently the dominant SOTA solutions due to their superior natural language understanding and generation capabilities.
>
> On certain specific numerical prediction tasks, traditionally optimized ML models can demonstrate performance comparable to or even superior to CatalystLLM. This phenomenon aligns with our expectations. Traditional ML models such as GAP-CatBERTa and GPTchem derive their advantage from being highly customized for single tasks and specific input formats. However, our target is not to surpass all highly specialized models on every single task, but rather to **develop a unified and general-purpose catalyst design assistant**. CatalystLLM's core value lies in its exceptional versatility and efficient integration capabilities. It can simultaneously handle 8 distinct task types within a unified framework, including information extraction, classification, numerical regression and open-ended semantic understanding, without requiring separate model design and training for each task.
>
> As shown in the table below, we present preliminary comparisons against multiple traditional ML baselines. For the Adsorption Prediction (AP), d-band Center Prediction (d-CP)  and Formation Prediction (FP) tasks, we introduce **mainstream graph neural network baseline models** currently prevalent in materials science, including CGCNN, SchNet, DimeNet++ and GemNet-OC. These models are widely regarded as standard benchmarks in adsorption energy prediction benchmarks such as OC20. We compare CatalystLLM against publicly available checkpoints of CGCNN, SchNet, DimeNet++ and GemNet-OC, all of which are trained on the same data set size as CatalystBench.
>
> | Model | AP | d-CP | FP |
> | :--- | :--- | :--- | :--- |
> | CGCNN | 0.79 | 0.66 | 0.78 |
> | SchNet | 0.72 | 0.65 | 0.77 |
> | DimeNet++ | 0.84 | 0.69 | 0.80 |
> | GemNet-OC | 0.85 | 0.72 | 0.84 |
> | CatBERTa | 0.82 | / | / |
> | GAP-CatBERTa | **0.86** | / | / |
> | GPTchem | / | 0.69 | **0.85** |
> | **CatalystLLM(ours)** | **0.81** | **0.73** | **0.80** |
>
> It is particularly noteworthy that these graph neural network models rely on **complete atomic 3D geometric coordinates** and lattice parameters as inputs. They explicitly encode 3D spatial structural information through message-passing mechanisms to achieve high-precision predictions. In contrast, CatalystLLM is limited to textual descriptions extracted from experimental literature or databases, along with simplified structured text sequences such as chemical formulas, crystal plane information and adsorption site descriptions.
>
> Experimental results demonstrate that CatalystLLM maintains competitive performance in adsorption energy prediction tasks. Its predictive accuracy falls slightly below that of top-tier graph models like GemNet-OC and DimeNet++, which rely on precise geometric structures. Moreover, CatalystLLM outperforms earlier SchNet models. This indicates CatalystLLM's ability to effectively capture chemical knowledge and structural features implicit in textual descriptions through multi-task fine-tuning. Crucially, widely-used GNNs require 3D coordinates derived from costly DFT relaxations. CatalystLLM achieves comparable accuracy using only text information. This enables instantaneous inference on vast chemical spaces where 3D structures are unknown.
>
> For the Faradaic Efficiency Prediction (FEP) task, we introduce traditional text prediction methods from the current scientific domain. As shown in the table below, although it falls short of specially trained state-of-the-art models for this task, CatalystLLM significantly outperforms other ML approaches, demonstrating considerable potential in this scientific domain.
>
> | Model | MSE | MAE | R² |
> | :--- | :--- | :--- | :--- |
> | multi-layer perceptron | 0.03 | 0.15 | 0.48 |
> | support vector regression | 0.06 | 0.20 | 0.11 |
> | linear regression | 0.05 | 0.17 | 0.20 |
> | BRR | 0.05 | 0.18 | 0.23 |
> | GPR | 0.04 | 0.15 | 0.34 |
> | Original paper | **0.01** | **0.08** | **0.84** |
> | **CatalystLLM(ours)** | **0.02** | **0.13** | **0.73** |
>
> We hope our detailed responses provide a clearer perspective on the model's performance and the dataset's construction. We are grateful for your thoughtful questions and welcome any further discussion to enhance the quality and impact of our study.

---

> ### Author Response · Authors · 2025-11-27
>
> Dear Reviewer,
>
> We would like to express our sincere gratitude once again for the time and professional effort you have dedicated to reviewing our manuscript. Your constructive comments have been instrumental in improving the quality of our work.
>
> We have previously submitted our detailed responses to your questions, along with the supplementary explanations and additional experimental results.
>
> As some time has passed since our rebuttal submission, we are writing to respectfully inquire about the status of the re-evaluation process. We fully appreciate the workload involved and respect the time required for a thorough review.
>
> If more time is needed for your deliberation, we completely understand and will patiently await your final assessment.
>
> Thank you again for your diligent work and support. We look forward to hearing from your valuable feedback.
>
> Sincerely.

---

### Official Review · Reviewer_SpDV · 2025-11-01

**Soundness:** 3
**Presentation:** 3
**Contribution:** 3
**Rating:** 6
**Confidence:** 4

**Summary:**

The paper proposes CatalystBench, a catalysis-focused benchmark that tries to cover the full catalyst R&D workflow in one place by assembling eight tasks: extracting catalyst materials and synthesis steps from literature, classifying the regulation strategy used (e.g., alloying, defect, interface modulation), predicting key catalytic properties from public datasets via natural-language instructions (adsorption energy, d-band center, formation/efficiency values), and understanding long “regulation-scheme” texts that explain why a given catalyst modification improves performance.

**Strengths:**

1. The paper correctly observes that catalysis papers mix narrative experimental text, semi-structured synthesis details, and simulation/numerical descriptors — and real catalyst design jumps between them. Existing benchmarks like ChemBench, LLM4Mat, SciBench are more fragmented and not tied to the full catalyst R&D loop. Their comparison table makes this point clearly.

2. The 8 tasks do cover the four capability buckets the authors call Understanding → Reasoning → Explaining (their Fig. 1 / Sec. 3.1) and they really do span text, structured catalyst descriptors, and numbers.

3.  Many “LLM-for-science” benchmarks just dump all tasks into SFT and hope; here they at least propose an architectural separation (MFT) and run ablations against other fine-tuning strategies (claimed in Sec. 3.2 / 3.3 and Appendix). That’s a reasonable, domain-motivated engineering idea.

**Weaknesses:**

1. A lot of the “hard” data is LLM-assisted before human filtering. For the regulation-scheme comprehension, the pipeline is: literature → GPT-4o (with SciQAG) → Q&A → human filtering. That is not the same as “fully human authored”, and it raises the usual questions: how much of the style/format is baked in from GPT-4o, how diverse the questions really are, and whether future GPT-4.x will get an unfair advantage because it’s closer to the generation distribution. The paper says experts “perform annotation and filtering,” but does not give inter-annotator agreement or rejection rates in the main text. I would ask to quantify that.

2. The biggest task, SE, has 6,612 instances; several key tasks (FEP 2,148; RMC 2,364; RSC 4,307) are in the low-thousands. The authors do mention that this “reflects a common constraint in catalytic research” and that they “prioritize task breadth”, but it does mean that (i) leaderboard variance could be high, and (ii) large multi-task LLMs may overfit the linguistic shell rather than the chemistry. This should be made more explicit.

3. MFT novelty is incremental. Multi-head, task-decoupled training for mixed classification/regression/generation is not new in ML; here it is justified by domain heterogeneity, but methodologically it’s a small step. For ICLR, I would like to see either (i) a clearer theoretical/optimization story (e.g., loss interference curves between numerical vs generative heads) or (ii) a stronger empirical claim like “without MFT, regression collapses by X%, with MFT, text tasks do not degrade.” Right now the paper mostly states superiority.

**Questions:**

How much is “real” experimental/literature text vs LLM-generated Q&A?

Are the 8 public catalytic datasets actually licensed / stable?

---

> ### Author Response · Authors · 2025-11-21
> **Answer (W1)**
>
> We sincerely thank you for your assessment of our work. We appreciate that you recognized the breadth of our benchmark task settings. Furthermore, we have developed a detailed revision plan addressing your comments.
>
> ### Weakness 1
> > A lot of the “hard” data is LLM-assisted before human filtering. For the regulation-scheme comprehension, the pipeline is: literature → GPT-4o (with SciQAG) → Q&A → human filtering....
>
>
>
> **R:** Thank you for this comment. Regarding your concerns about stylistic bias, future model advantages and quantitative metrics, we respond as follows:
>
> **1. Mitigation of Stylistic Bias**
> To mitigate stylistic bias in open-ended Q&A pairs, we implement the following mitigation efforts:
> *   **Multi-round Generation:** First, during the generation phase, we avoid direct repetition of single-round outputs. As detailed in **Appendix A.5.2 OPEN-ENDED Q&A DATA CALIBRATION**, we design a multi-round generation process. The model first extracts keywords and crucial reaction mechanisms from literature, then guides the model to generate logical chains focused on core scientific concepts. Beyond the details presented in **Appendix E.1 PROMPTS FOR Q&A GENERATING**, our prompts additionally incorporate variations to guide the model in altering sentence structures. These include switching between active and passive voice, rearranging subordinate clauses and prohibiting bullet-point responses to ensure linguistic diversity. This process helps ensure that the final Q&A pairs reflect the scientific logic of the catalysis field rather than the linguistic habits of GPT-4o.
> *   **Multi-round Evaluation:** Furthermore, we observe that open-ended Q&A pairs are significantly influenced by the output style of LLMs. In such scenarios, answers exhibiting similar linguistic styles may receive higher evaluation scores even when containing factual errors. Therefore, we design a multi-round evaluation process, as detailed in **Appendix E.3 PROMPTS FOR LLM-BASED EVALUATION**. We instruct the evaluation LLM to extract keywords without altering answer content or semantics, then compare based on the extracted semantic information. This further mitigates bias issues arising from LLM-assisted generation of open-ended Q&A pairs.
>
> **2. Addressing Future GPT-4.x Advantage**
> Regarding the risk of GPT-4.x gaining an unfair advantage based on distributional proximity in this benchmark, we offer the following clarifications:
> 1.  First, the core challenge of CatalystBench still lies in scientific reasoning and factual accuracy, not in matching specific linguistic styles. Our evaluation involves multiple rounds of design refinement, keyword extraction and content distillation while preserving original meaning. Even if future GPT-4.x models maintain consistent response styles, they cannot achieve high scores on plausibility metrics without accurately answering expert-verified complex scientific facts.
> 2.  Additionally, we incorporate expert participation in both generation filtering and evaluation to avoid complete reliance on the LLM's own generation and assessment capabilities. **Appendix C.2 EXPERT EVALUATION** compares the consistency between LLM scores and expert ratings, revealing a Pearson correlation coefficient of **0.89**. This indicates that the two evaluation systems closely align in assessing overall answer quality trends.
> 3.  Finally, only the Regulation Scheme Comprehension (RSC) task heavily relies on text generation. Most benchmark tasks utilize structured data templates or numerical regression, where “stylistic bias” effects can be disregarded.
>
> **3. Quantitative Metrics**
> To meet quantitative requirements of the expert annotation process, we will supplement the following data in the paper:
> 1.  **Rejection rate:** As we illustrate at the end of **Appendix A.5.2 OPEN-ENDED Q&A DATA CALIBRATION**, approximately **9.2%** of generated samples are rejected during expert screening due to hallucinations or lack of scientific depth. Specific erroneous samples are detailed in **Appendix A.6 ERROR SAMPLES**.
> 2.  **Inter-Annotator Agreement (IAA) [New Experiment]:** To further validate the rigor of our filtering process, we conduct IAA analysis on 500 randomly selected samples. Two domain experts independently annotate each sample as accept or reject achieving a **Cohen's Kappa coefficient of 0.75**. This further confirms that the dataset reflects domain knowledge rather than model bias. The expert rejection rate for new samples was **6.8%, closely matching previous results**. This demonstrates the overall stability and reliability of the expert annotation setup.

---

> ### Author Response · Authors · 2025-11-21
> **Answer (W2)**
>
> ### Weakness 2
> > The biggest task, SE, has 6,612 instances; several key tasks (FEP 2,148; RMC 2,364; RSC 4,307) are in the low-thousands. The authors do mention that this “reflects a common constraint in catalytic research” and that they “prioritize task breadth”, but it does mean that (i) leaderboard variance could be high, and (ii) large multi-task LLMs may overfit the linguistic shell rather than the chemistry. This should be made more explicit.
>
>
>
> **R:** Thank you for this comment. This comment accurately identifies a core limitation in catalytic science research stemming from data scale, which is crucial for refining our existing work. This reflects a major challenge in applying AI to catalysis: the high-fidelity and experimentally validated data is far scarcer than general text data. Regarding the two specific issues you raised, we offer the following explanations:
>
> **1. Leaderboard variance could be high.**
> On small datasets, random fluctuations in model performance can indeed affect the reliability of evaluation rankings. To mitigate this issue and provide more robust assessments, we implement the following measures:
> 1.  First, in **Sec 6.1 COMPARISON OF FINE-TUNING STRATEGIES**, we plot error bars for all fine-tuning strategies' performance comparisons in Fig 5. These represent the mean and standard deviation calculated from 5 independent evaluations. For LLM evaluation in **Sec 6.2 MAIN BENCHMARK RESULTS**, we report MAE score across all numerical prediction tasks in the main results Table 1, which partially compensates for instability arising from limited data volume.
> 2.  Additionally, we conduct ablation experiments and cross-task generalization assessments for different strategies in **Appendix B.5 ADDITIONAL ABLATION STUDY**, indirectly reflecting model robustness under varying data configurations such as different weighted loss-function.
>
> **2. Multi-task LLMs may overfit the linguistic shell rather than the chemistry.**
> With limited data, LLMs indeed risk overfitting linguistic patterns instead of learning underlying scientific principles. We design relevant experiments and analyses in our research to mitigate this issue:
> 1.  **The MFT architecture** itself enables the model to learn cross-task heterogeneous chemical features within a shared backbone network by employing independent output heads for classification, regression and generation tasks. And we use Low-Rank Adaptive (LoRA) instead of full parameter fine-tuning. By freezing most parameters of the pre-trained ChemLLM and training only a small number of low-rank adapters, we significantly limit the model's ability to memorize specific training samples or linguistic features.
> 2.  Furthermore, the **Sec 6.3 ABLATION STUDY** separately compares **the impact of cross-task testing and different input formats** on results. **Sec 6.3.1 COLLABORATIVE EFFECTS OF TASK COMBINATIONS** demonstrates that removing training data for one task affects performance on other related tasks. This indicates the model does not learn each task in isolation but acquires a unified and interconnected knowledge system of catalysis. **Sec 6.3.2 THE IMPACT OF INPUT FORMAT ON MODEL PERFORMANCE** demonstrates that transitioning inputs from simple SMILES strings to explicit structured descriptions including active sites, coordination environments and adsorbates significantly improves model performance across all prediction tasks. This indicates that CatalystLLM's performance ceiling depends on the density and explicitness of chemical information in inputs, without overfitting linguistic descriptions.
> 3.  Finally, **Appendices C.2 and C.3** demonstrate CatalystLLM's genuine chemical reasoning capability through **expert evaluations**. While general LLMs generate linguistically fluent responses, domain experts prefer CatalystLLM's outputs, whose quality relies on the accuracy of its chemical knowledge.
>
> In summary, through multidimensional experiments and a chain of evidence, we demonstrate that CatalystLLM can learn the **underlying principles of catalytic chemistry** to a certain extent. We acknowledge that fundamentally expanding the data scale for each task represents a future direction. During this initial dataset development phase, we strive to provide the most robust evaluation and analysis possible within the CatalystBench framework. In future iterations, we will further expand both the scale of data and the breadth of tasks, thereby providing the field with more comprehensive and meticulous benchmarking.

---

> ### Author Response · Authors · 2025-11-21
> **Answer(W3)**
>
> ### Weakness 3
> > MFT novelty is incremental. Multi-head, task-decoupled training for mixed classification/regression/generation is not new in ML...
>
>
>
> **R:** Thank you for this comment. We acknowledge that multi-head architectures have been explored in general machine learning domains, but our core contribution lies in experimentally validating and optimizing the MFT strategy for the unique challenges of task heterogeneity in catalytic science. In response to your request, we will provide clearer analysis through theoretical and quantitative experiments.
>
> **1. Theoretical analysis**
>
> In catalytic science, different tasks exhibit inherent heterogeneity. **Generation tasks** such as regulation scheme comprehension operate in discrete token spaces and learn semantic logic by optimizing cross-entropy loss. In contrast, **regression tasks** like adsorption energy prediction reside in continuous numerical spaces and require the model to perform high-precision floating-point mapping. Prediction tasks strictly adhere to physicochemical laws and typically aim to minimize mean squared error. In standard full-task fine-tuning frameworks, forcing the model to predict high-precision numerical values like “-1.6033 eV” through token classification creates a modality gap. The **gradient direction** for minimizing token perplexity often conflicts with the gradient for reducing numerical error, resulting in destructive interference.
>
> The Regression tasks often suffer disproportionate harm for being highly sensitive to numerical variations. To address this, our proposed MFT strategy theoretically **decouples optimization objectives**. The model learns general chemical representations through a shared backbone network, while decoupled task-specific heads project these representations into discrete and continuous task manifolds, thereby effectively orthogonalizing the loss space.
>
> **2. Quantitative experiments**
>
> To validate this conclusion, we conduct quantitative analysis as suggested.
>
> First, we reanalyze **the original experimental data** in Fig 5 to quantify the necessity of MFT method, as shown below. We find that for 4 regression prediction tasks, the Base-FT strategy achieves an average R² improvement of 12.88% over the Base-ST strategy. For 5 additional text tasks, the Base-FT strategy achieves an average accuracy improvement of 5.6% over the Base-ST strategy. This indicates that multi-task fine-tuning delivers particularly significant improvements in regression tasks while maintaining advantages in text tasks to some extent. However, the performance gains from current strategies **fall short of achieving optimal levels** across all tasks. Theoretical analysis suggests that model potential is constrained when continuous regression values and discrete text tokens are forced to share the same projection head.
>
> | | ME | SE | RMC | FEP | AP | d-CP | FP | RSC |
> | :--- | :---: | :---: | :---: | :---: | :---: | :---: | :---: | :---: |
> | **Task-specific** | 0.95 | 0.86 | 0.71 | 0.57 | 0.69 | 0.61 | 0.71 | 0.75 |
> | **FT strategy** | 0.98 | 0.88 | 0.79 | 0.67 | 0.78 | 0.70 | 0.76 | 0.76 |
> | **MFT strategy** | 0.98 | 0.89 | 0.81 | 0.73 | 0.81 | 0.73 | 0.80 | 0.79 |
>
> We then quantify the marginal benefit of the MFT strategy's decoupled output heads compared to Base-FT method.
> *   For 4 regression prediction tasks, the MFT strategy achieves an average **R² improvement of 4.1%** over the Base-FT strategy.
> *   For 5 text tasks, the MFT strategy achieves an average **Accuracy improvement of 1.8%** over the Base-FT strategy.
>
> This demonstrates that the multi-task framework with decoupled output heads enables regression tasks to fully leverage shared representations. Meanwhile, text tasks maintain stable performance with slight improvements.
>
> **3. Gradient Analysis**
>
> Additionally, we analyze the gradients propagating from different heads to the shared LLM backbone layer within the MFT strategy, validating the effectiveness of the MFT framework. We compute the cosine similarity of gradients between regression heads and generation heads in the final hidden layer during training. The **gradient cosine similarity distribution** predominantly remained non-negative, with an average value of **approximately 0.12**. The similarity of ~0.12 indicates that the tasks are largely orthogonal, and the positive sign suggests a slight synergistic effect, which is the ideal scenario for Multi-Task Learning. This indicates that MFT effectively decouples task-specific noise. The mean squared error loss for numerical precision does not interfere with the linguistic features required for text generation. Concurrently, positive gains are observed between gradients. This signifies that the shared representations learned by the backbone network benefit both tasks simultaneously.
>
> In summary, this strategy demonstrates the importance of decoupling tasks for complex scientific discovery tasks that simultaneously involve numerical reasoning and text generation.

---

> ### Author Response · Authors · 2025-11-21
> **Answer (Q1 & Q2)**
>
> ### Question 1
> > How much is “real” experimental/literature text vs LLM-generated Q&A?
>
> **R:** Thank you for this question. The clear categorization of tasks in Table 3 of **Appendix A DATA CONSTRUCTION** is explained as follows:
> 1.  **“Real” experimental/literature text (approx. 84%):** 7 out of 8 task categories including ME, SE, RMC, FEP, AP, d-CP and FP tasks comprise a total of 22,604 instances. These are directly sourced from scientific literature or datasets. They are generated using fixed templates. Their content origin is entirely authentic, while their format employs template-based processing, incorporating predefined styles and formats.
> 2.  **LLM-generated Q&A Data (approx. 16%):** Only the RSC task involves LLM-assisted generation followed by human filtering, achieving a 90.8% success rate in manual review.
>
> ---
>
> ### Question 2
> > Are the 8 public catalytic datasets actually licensed / stable?
>
>
>
> **R:** Thank you for this question. All 8 publicly available catalyst datasets used in this study **possess valid licenses**. The license types for each dataset are detailed in the table below. We confirm that the usage in this study fully complies with the academic research purposes permitted by all original license agreements.
>
> | Dataset | License |
> | :--- | :--- |
> | OC20 | CC-BY 4.0 |
> | Materials Project | CC-BY 4.0 |
> | Other datasets | MIT or CC-BY license |
>
> Regarding dataset stability, some key datasets originate from published research literature with corresponding DOI identifiers for direct traceability. Furthermore, core datasets including OC20, the Materials Project and ElectroCatalytic Reduction datasets are hosted in institutional repositories committed to long-term maintenance, ensuring data accessibility and stability.
>
> We thank you for these crucial observations. We hope our clarifications alleviate your concerns regarding data scale and potential overfitting. Your feedback is invaluable, and we look forward to advancing the ongoing refinement of this work through our discussions.

---

> ### Author Response · Authors · 2025-11-27
>
> Dear Reviewer,
>
> We would like to express our sincere gratitude once again for the time and professional effort you have dedicated to reviewing our manuscript. Your constructive comments have been instrumental in improving the quality of our work.
>
> We have previously submitted our detailed responses to your questions, along with the supplementary explanations and additional experimental results.
>
> As some time has passed since our rebuttal submission, we are writing to respectfully inquire about the status of the re-evaluation process. We fully appreciate the workload involved and respect the time required for a thorough review.
>
> If more time is needed for your deliberation, we completely understand and will patiently await your final assessment.
>
> Thank you again for your diligent work and support. We look forward to hearing from your valuable feedback.
>
> Sincerely.

---

### Official Review · Reviewer_UAyg · 2025-11-03

**Soundness:** 2
**Presentation:** 3
**Contribution:** 2
**Rating:** 4
**Confidence:** 3

**Summary:**

This paper presents CatalystBench, a large-scale multi-task benchmark for evaluating language models in catalysis science. It covers seven task categories, from catalyst knowledge to mechanism reasoning and molecular design, built through automated data extraction and expert validation. Experiments across general and chemistry-specific LLMs reveal that while models handle basic factual tasks reasonably well, they still perform poorly on mechanism-level reasoning and condition prediction, highlighting major gaps in domain understanding.

**Strengths:**

1. Evaluating LLMs in catalysis is both scientifically important and practically valuable. The benchmark spans multiple levels of reasoning, from basic recognition to scientific inference.
2. The combination of automated extraction and expert validation ensures data reliability. Figures and examples make the complex tasks easy to follow.

**Weaknesses:**

1. Although the results highlight model weaknesses, the paper does not analyze why models fail (e.g., due to missing domain knowledge, poor reasoning ability, or lack of grounding). Adding this discussion would make the findings more actionable.
2. The paper could include more analysis of data quality and diversity, such as the balance of catalyst types, reaction categories, and distribution of task difficulties.
3. While the benchmark is comprehensive, the work is mainly a dataset contribution, with relatively limited technical novelty in terms of modeling or learning methodology.

**Questions:**

Please refer to the cons

---

> ### Author Response · Authors · 2025-11-21
> **Answer(W2 & Q2):**
>
> We sincerely appreciate your review of the manuscript and the constructive feedback provided. Thank you for acknowledging the comprehensiveness of our benchmarking approach. Furthermore, we plan to revise the manuscript thoroughly addressing your comments.
>
> Due to space constraints, we have placed the response to weakness 2 at the beginning. We apologise for any inconvenience caused.
>
> ### Weakness 2 & Question 2
> > The paper could include more analysis of data quality and diversity, such as the balance of catalyst types, reaction categories, and distribution of task difficulties.
>
> **R:** Thank you for this comment. We agree that a more detailed analysis of the dataset quality and diversity is crucial. We have added a detailed statistical analysis and visualization of CatalystBench, including the distribution of catalyst types, the proportion of catalyst regulation scheme categories and the distribution of reaction types. We will add them into supplementary content.
>
> For dataset analysis, we compiled key metal elements in catalysts across the dataset, particularly in tasks derived from experimental literature such as ME, SE and RMC tasks. As shown in the table below, Cu is the most frequently occurring element, accounting for 43.7%. This aligns with our data sources' focus on the current state of research in CO₂ electrocatalytic reduction (CO₂RR), where copper-based catalysts represent a major research focus. Simultaneously, the dataset contains largely transition metals such as Fe, Co and Ni, which play crucial roles in electrocatalysis, along with various other metallic elements. This ensures the model can learn diverse elemental knowledge.
>
> | Key metal elements | Count | Percentage |
> | :--- | :--- | :--- |
> | Cu | 6,888 | 43.7% |
> | Fe | 3,279 | 20.8% |
> | Ni | 2,412 | 15.3% |
> | Co | 1,497 | 9.5% |
> | Others (Pd, In, Bi, etc.) | 1,671 | 10.6% |
>
> CatalystBench incorporates a variety of advanced catalyst design and control strategies. We statistically analyze 2,364 samples from the RMC task. As shown in the table below, Structure control and Composite materials are the two most prevalent approaches, accounting for 46.5% and 22.3% respectively. This reflects the current mainstream trend of enhancing catalytic performance through constructing heterojunctions and multi-component synergies. Additionally, Surface modification, Alloying, Defect engineering and Atomic-level dispersion also account for significant proportions, comprehensively covering key technologies from macroscopic morphology to atomic-scale regulation.
>
> | Regulation Method Type | Count | Percentage |
> | :--- | :--- | :--- |
> | Structure Control | 1,099 | 46.5% |
> | Composite | 527 | 22.3% |
> | Atomic Level Dispersion | 235 | 9.9% |
> | Surface/interface Modification | 211 | 8.9% |
> | Alloy | 182 | 7.7% |
> | Defect | 110 | 4.7% |
>
> CatalystBench also encompasses several core reaction types in the electrocatalysis field. As shown in the table below, CO₂ electrocatalytic reduction (CO₂RR) constitute the largest proportion of the dataset, accounting for 64.3%, directly attributable to our integration of multiple literature sources and datasets focused on CO₂RR. Simultaneously, by incorporating datasets like SACs and OC20, we have also included a substantial amount of theoretical calculation data related to oxygen evolution reaction (OER) and oxygen reduction reaction (ORR), combined as 22.6%. Additionally, a small number of other surface adsorption and catalytic reactions are included, collectively forming a diverse set of reaction scenarios.
>
> | Reaction Type | Percentage |
> | :--- | :--- |
> | CO₂RR | 64.3% |
> | OER/ORR | 22.6% |
> | Others (surface adsorption, etc.) | 13.1% |
>
> This detailed statistical analysis will enhance the transparency and useability of CatalystBench, providing deeper insights into its strengths, current coverage and potential biases, facilitating in-depth applications of this benchmark in the future.

---

> ### Author Response · Authors · 2025-11-21
> **Answer(W1 & Q1)**
>
> ### Weakness 1 & Question 1
> > Although the results highlight model weaknesses, the paper does not analyze why models fail (e.g., due to missing domain knowledge, poor reasoning ability, or lack of grounding). ...
>
>
>
> **R:** Thank you for this comment. We agree that analyzing the reasons for model failure is important. We have conducted a systematic analysis to thoroughly examine the limitations of CatalystLLM and other superior LLMs across 3 dimensions of LLM capabilities in materials science: Understanding, Reasoning and Explaining.
>
> **1. Understanding Failure Analysis**
>
> LLM comprehension is primarily demonstrated through information extraction and classification tasks in scientific texts. Our analysis of confusion patterns reveals that errors predominantly stem from classification ambiguity and context span limitations. We collected error samples for statistical analysis by domain experts participating in the evaluation.
>
> | Task Type | Primary Error Type | Frequency | Reason of Failure |
> | :--- | :--- | :--- | :--- |
> | **SE** | Omission Error | ~65% of errors | **Long-Context Forgetting:** The model fails to attend to fine-grained details in lengthy synthesis procedures. For instance, it may omit complex conditional attributes of certain operations.|
> | **RMC** | Boundary Confusion | ~72% of errors | **Taxonomic Ambiguity:** The current regulation scheme classifier reveals conceptual overlap between some classes from scientific literature, such as between “defect” and “surface modification”. When defects are surface-induced, semantic boundaries become blurred, making it difficult for models to discern their primary label. |
>
> **2. Reasoning Failure Analysis**
>
> LLM reasoning capability effects in performance prediction based on catalytic material texts or chemical descriptions. Taking catalyst adsorption energy prediction as an example, we analyze high-error samples accounting for 19% residual variance. Comparing error types, we find that the errors are from a few structural types. As shown in the table below, failures have two fundamental causes:
>
> 1.  **Modality Gap:** Although CatalystLLM efficiently processes text or SMILES representations, these one-modal characterizations fail to explicitly capture 3D steric hindrance within complex amorphous structures.
>
> 2.  **Data Sparsity:** The model exhibits larger error rates for catalysts containing rare elements, indicating insufficient domain-specific knowledge for this particular subset.
>
> | Catalyst Category | Examples | Structural Complexity | Failure Type | Root Cause Analysis |
> | :--- | :--- | :--- | :--- | :--- |
> | **Simple Surfaces** | Pt(111), Cu(100) | Low | Minimal Error (MAE < 0.15 eV) | Text descriptions sufficiently define the active site environment. The model successfully maps descriptors to energy. |
> | **High-Entropy Alloys** | Ir-Rh-Ru complexes | Medium/High | Knowledge Gap | **Data Sparsity:** Performance affected by element frequency in training data. For rare element combinations, LLM parameters fail to learn robust relationships. |
> | **Amorphous Clusters** | Defective Nanoclusters | Extreme | Reasoning Failure (High Variance) | **Modality Gap:** The inputs cannot capture 3D steric effects or coordinate geometry. LLM fails to reason about spatial constraints without graph inputs. |
>
> **3. Explaining Failure Analysis**
>
> We further classify errors in the Regulation Scheme Comprehension (RSC) task into knowledge hallucinations and logical fallacies, as shown in the table below. We specifically analyze error samples listed in **Appendix C.3**.
>
> | Error Type | Description | Specific Case Study & Analysis |
> | :--- | :--- | :--- |
> | **Hallucination** | The model generates schemes that contradict established physicochemical principles. | **Case:** Zn sites in Cu-Zn bimetallic system. |
> | | | **Model Claim:** “Zn sites directly participate in C-O bond cleavage.” |
> | | | **Scientific Truth:** Zn sites stabilize intermediates via electrostatic fields. |
> | | | **Analysis:** The model hallucinates a mechanism based on generic “active site” priors rather than the specific “tandem catalysis” context, showing a lack of grounding. |
> | **Logical Fallacy** | The model correctly identifies components but fails to derive the correct causal outcome. | **Case:** N-doped graphene for Fe single-atom. |
> | | | **Model Claim:** Attributes performance solely to “enhanced conductivity.” |
> | | | **Scientific Truth:** The primary driver is the modulation of the electronic structure. |
> | | | **Analysis:** The model is biased to a generic explanation instead of reasoning through the specific electronic interaction induced by N-doping. |
>
> We believe these additions enhances the validity of the benchmark results, directing to future works for improving LLMs ability for scientific fields. For instance, in tasks such as adsorption energy prediction, multimodal LLMs can integrate textual information with 3D structural data to improve the reasoning reliability.

---

> ### Author Response · Authors · 2025-11-21
> **Answer (W3 & Q3)**
>
> ### Weakness 3 & Question 3
> > While the benchmark is comprehensive, the work is mainly a dataset contribution, with relatively limited technical novelty in terms of modeling or learning methodology.
>
>
>
> **R:** Thank you for this comment and for acknowledging the comprehensiveness of our benchmark. We believe that CatalystBench itself represents significant contribution to the field to drive more systematic and reproducible research.
>
> Beyond our benchmarking contribution, we wish to elaborate on our understanding of **the methodological innovation** proposed in the paper. We acknowledge that multi-head architectures are not innovative in the broad machine learning field. However, our contribution lies not in the architecture itself, but in its application to address newly identified and domain-specific challenges in catalytic science, coupled with systematic validation. Specifically:
>
> 1.  **Identifying critical domain-specific problems:** By constructing CatalystBench, we systematically reveal for the first time the comprehensive and heterogeneous task landscape in catalytic research and discovery. This workflow contains 3 task categories: tasks requiring high-precision numerical regression such as adsorption energy calculation, categorical classification tasks such as method classification, and open-ended scientific reasoning tasks. Traditional multi-task fine-tuning approaches often fail in this scenario due to severe interference from conflicting objectives.
>
> 2.  **Providing the first empirical solution:** This study delivers the first systematic empirical evaluation in the catalysis domain, demonstrating that the decoupled Multi-head Full-task (MFT) architecture, specifically designed for multiple heterogeneous tasks in catalysis domain, works effectively. As demonstrated in Fig 5 and subsequent analyses, the MFT strategy achieves certain improvements over standard fine-tuning. For 4 regression prediction tasks, the MFT strategy achieves an average R² improvement of 4.1% over the Base-FT strategy. For 5 text tasks, the MFT strategy achieves an average F1 improvement of 1.8% over the Base-FT strategy.
>
> Thus, we summarize our core contributions in two aspects:
>
> 1.  **Foundational Data Contribution.** We provide the comprehensive and in-depth CatalystBench to advance domain standardization and future research.
> 2.  **Practical Methodological Contribution.** We identify critical challenges faced by LLMs in the catalysis domain and validates a robust and effective strategy to address them, laying a promising baseline for future research.
>
> We appreciate your feedback and will elaborate the introduction and discussion sections in the final revision to more clearly articulate the above contribution framework. We sincerely hope our responses have adequately addressed your concerns. We thank you again for your insightful comments, which have significantly improved our manuscript, and we look forward to further discussions with you to continually refine this work.

---

> ### Author Response · Authors · 2025-11-27
>
> Dear Reviewer,
>
> We would like to express our sincere gratitude once again for the time and professional effort you have dedicated to reviewing our manuscript. Your constructive comments have been instrumental in improving the quality of our work.
>
> We have previously submitted our detailed responses to your questions, along with the supplementary explanations and additional experimental results.
>
> As some time has passed since our rebuttal submission, we are writing to respectfully inquire about the status of the re-evaluation process. We fully appreciate the workload involved and respect the time required for a thorough review.
>
> If more time is needed for your deliberation, we completely understand and will patiently await your final assessment.
>
> Thank you again for your diligent work and support. We look forward to hearing from your valuable feedback.
>
> Sincerely.

---

### Author Response · Authors · 2025-12-02
**The Summary of 11014 Rebuttal**

Dear Area Chair,

We are the authors of the Submission11014. We sincerely appreciate your time and effort in managing the review process for our submission. Given the recent technical issues with OpenReview and the assignment of a new AC, we would like to provide a concise summary of our rebuttal progress and the status of reviewer interactions.

### Status of Reviewer Interactions & System Outage

Our paper received **initial scores of 8, 6, 6 and 4**. During the rebuttal, we provided detailed responses and new experimental results to all reviewers.

**Reviewer 2hjb (Score: 8)**: On 28 Nov 2025, 01:12, this reviewer explicitly responded to our rebuttal, stating that all their concerns had been fully resolved.

**Other Reviewers**: We sent reminders to the other reviewers on 27 Nov 2025, 23:53. Unfortunately, due to the subsequent OpenReview system outage and the closing of the discussion period, they were unable to submit their feedback or update their scores in the system.


### Summary of Key Rebuttal Improvements
We have addressed the reviewers’ concerns thoroughly with new experiments and analyses. Below is a summary of the major updates:

**- Comparison with SOTA Graph Neural Networks (Addressing Reviewer 3 & Reviewer 4):**

Reviewers 3 and 4 questioned the performance compared to geometry-aware models. We added comparisons against state-of-the-art GNNs including GemNet-OC, DimeNet++ and SchNet on adsorption energy prediction tasks.

**Result:** CatalystLLM achieves comparable performance ($R^2=0.81$ vs. GemNet-OC$R^2=0.85$) **using only text inputs**, without requiring the costly 3D geometric coordinates that GNNs depend on. This highlights the model’s practical value in screening vast chemical spaces where 3D structures are unknown.

**- Systematic Failure Analysis (Addressing Reviewer 1 & Reviewer 4):**

We conducted a comprehensive failure analysis across three dimensions: Understanding, Reasoning, and Explaining.

**Analysis:** We identified that “Reasoning Failures” (19% residual variance) primarily stem from the modality gap in processing complex amorphous clusters and data sparsity in high-entropy alloys. We also analyzed “Understanding Failures” caused by taxonomic ambiguity and “Explaining Failures” caused by hallucinations in open-ended tasks.

**-Theoretical & Empirical Validation of MFT Strategy (Addressing Reviewer 1, Reviewer 2 & Reviewer 4):**

To address concerns about the novelty and theory of our Multi-head Full-task (MFT) architecture:

**Theoretical:** We calculated the gradient cosine similarity ($\sim 0.12$) between regression and generation heads, quantitatively proving that our architecture effectively orthogonizes the loss space to prevent gradient conflict while maintaining positive transfer.

**Empirical:** We added ablation studies showing MFT provides a 4.1% $R^2$ improvement in regression tasks compared to standard fine-tuning.

**- Data Quality and Rigor (Addressing Reviewer 1 & Reviewer 2):**

We provided detailed statistics on elemental distribution and reaction types. Additionally, we conducted an Inter-Annotator Agreement (IAA) study (Cohen’s Kappa = 0.75) to validate the reliability of our expert-filtered dataset, mitigating concerns about LLM-generated content bias.

### Conclusion
We believe our extensive new experiments and analyses have solidly addressed the critiques regarding baselines, failure analysis, and methodological validation. We respectfully request that you consider these updates and the positive feedback from the responding reviewer when making your final decision.

Thank you again for your service to the community. Please feel free to reply and discuss any questions you may have, and we will respond as soon as possible. We sincerely appreciate you taking the time to review our situation amidst your busy schedule and we truly recognize the heavy workload and effort involved in your role. Thank you very much for your time and dedication.

Best regards,

The Authors

---

### Meta-Review · Area_Chair_3rqW · 2025-12-24

**Summary:**

This submission introduces CatalystBench, a multi-task benchmark intended to reflect a catalyst R&D workflow, and a domain fine-tuning approach (MFT) with decoupled heads for heterogeneous tasks.

Reviewers agreed the benchmark scope and workflow alignment are valuable, and the empirical results suggest meaningful gains for a unified catalyst assistant.

The main concerns were (1) rigor and potential bias from LLM-assisted data generation and evaluation, including whether the benchmark distribution could advantage related models, (2) limited analysis of dataset coverage, diversity, and potential skew toward CO$_{2}$RR and electrocatalysis, (3) limited technical novelty of MFT relative to prior multi-head multi-task training, with an earlier lack of theoretical or optimization evidence, and (4) missing or insufficient comparisons to strong non-LLM baselines for numerical tasks (notably geometry-aware GNNs), plus limited failure analysis explaining residual errors.

**Reviewer Concerns:**

**Addressed by the rebuttal:**

- Missing strong regression baselines: the authors added comparisons to standard materials GNNs (SchNet, DimeNet++, GemNet-OC, etc.) and clarified the text-only setting relative to geometry-dependent methods.
- Lack of failure analysis: the authors added structured error analyses across understanding, reasoning, and explaining, including a breakdown of AP residual variance with identified high-error regimes.
- Data quality and diversity analysis: the authors provided additional dataset statistics (element distribution, regulation method types, reaction type mix) and reported an inter-annotator agreement study (Cohen’s kappa), plus an expert-filter rejection rate.
- Lack of theoretical support for MFT: the authors added a gradient cosine similarity analysis between heads to argue reduced gradient conflict.

**Still partially outstanding:**

- Scope and bias: the benchmark remains heavily centered on electrocatalysis and CO$_{2}$RR. Expansion is described as future work rather than resolved in this version.
- LLM-assisted pipeline risks: while mitigation steps and IAA help, questions remain about distributional bias and evaluation coupling in the generation-heavy task(s). Key details should be clearly surfaced in the main paper and accompanied by a reproducible protocol.
- Novelty: even with added analyses, MFT remains an incremental methodological contribution. The strongest contribution is the benchmark and systematic evaluation.

**Reviewer Scores:**

Reviewer SpDV (6): Likely unchanged at 6. Added IAA, rejection rate, and GNN baselines address key questions, but concerns about dataset scale, potential distributional bias, and incremental method novelty remain. This still reads as borderline rather than clearly strengthened to an 8.

Reviewer 2hjb (8): Likely unchanged at 8. This reviewer explicitly stated their concerns were resolved after the rebuttal.

Reviewer dAYs (6): Likely unchanged at 6. The rebuttal partially improves baseline breadth and acknowledges coverage limits, but does not fundamentally change the scope limitation or move the work beyond a solid benchmark plus a reasonable engineering baseline.

Reviewer UAyg (4): Likely unchanged at 4, or at most a small shift toward the border. The rebuttal adds the missing failure analysis and additional dataset statistics, which improves actionability, but the reviewer’s core view (dataset-centric contribution with limited modeling novelty) likely remains. Under the stated guideline, this is not clearly strong enough to justify a definite upgrade.

---

### Decision · Program_Chairs · 2026-01-26

Accept (Poster)